# Boosting Unsupervised Semantic Segmentation with Principal Mask Proposals

**Oliver Hahn**[1]     **Nikita Araslanov**[2,3]     **Simone Schaub-Meyer**[1,4]     **Stefan Roth**[1,4]
[1] *Department of Computer Science, TU Darmstadt*     [2] *Department of Computer Science, TU Munich*
[3] *Munich Center for Machine Learning (MCML)*     [4] *hessian.AI*

**Reviewed on OpenReview:** *https://openreview.net/forum?id=UawaTQzfwy*

## Abstract

Unsupervised semantic segmentation aims to automatically partition images into semantically meaningful regions by identifying global semantic categories within an image corpus without any form of annotation. Building upon recent advances in self-supervised representation learning, we focus on how to leverage these large pre-trained models for the downstream task of unsupervised segmentation. We present PriMaPs – Principal Mask Proposals – decomposing images into semantically meaningful masks based on their feature representation. This allows us to realize unsupervised semantic segmentation by fitting class prototypes to PriMaPs with a stochastic expectation-maximization algorithm, PriMaPs-EM. Despite its conceptual simplicity, PriMaPs-EM leads to competitive results across various pre-trained backbone models, including DINO and DINOv2, and across different datasets, such as Cityscapes, COCO-Stuff, and Potsdam-3. Importantly, PriMaPs-EM is able to boost results when applied orthogonally to current state-of-the-art unsupervised semantic segmentation pipelines. Code is available at `https://github.com/visinf/primaps`.

## 1 Introduction

Semantic image segmentation is a dense prediction task that classifies image pixels into categories from a pre-defined semantic taxonomy. Owing to its fundamental nature, semantic segmentation has a broad range of applications, such as image editing, medical imaging, robotics, or autonomous driving (see Minaee et al., 2022, for an overview). Addressing this problem via supervised learning requires ground-truth labels for every pixel (Long et al., 2015; Ronneberger et al., 2015; Chen et al., 2018b). Such manual annotation is extremely time and resource intensive. For instance, a trained human annotator requires an average of 90 minutes to label up to 30 classes in a single 2 MP image (Cordts et al., 2016). While committing significant resources to large-scale annotation efforts achieves excellent results (Kirillov et al., 2023), there is natural interest in a more economical approach. Alternative lines of research aim to solve the problem using cheaper – so-called "weaker" – variants of annotation. For example, image-level supervision describing the semantic categories present in the image, or bounding-box annotations, can reach impressive levels of segmentation accuracy (Dai et al., 2015; Araslanov & Roth, 2020; Oh et al., 2021; Xu et al., 2022; Ru et al., 2023).

As an extreme problem scenario toward reducing the annotation effort, unsupervised semantic segmentation aims to consistently discover and categorize image regions in a given data domain without any labels, knowing only how many classes to discover. Unsupervised semantic segmentation is highly ambiguous as class boundaries and the level of categorical granularity are task-dependent.[1] However, we can leverage the fact that typical image datasets have a homogeneous underlying taxonomy and exhibit invariant domain characteristics. Therefore, it is still feasible to decompose images in such datasets in a semantically meaningful and consistent manner without annotations.

---

[1]While assigning actual semantic labels to regions without annotation is generally infeasible, the assumption is that the categories of the discovered segments will strongly correlate with human notions of semantic meaning.

| Image | Mask 1 | Mask 2 | Mask 3 | PriMaPs *(all)* | Pseudo Label |
|---|---|---|---|---|---|

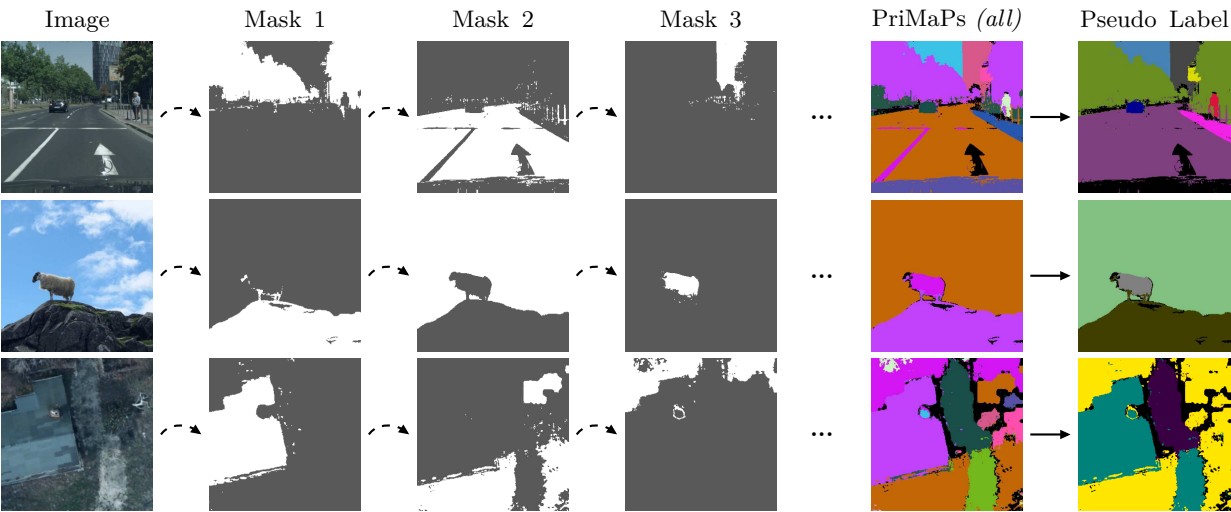

Figure 1: **PriMaPs pseudo-label example.** Principal mask proposals (PriMaPs) are iteratively extracted from an image (dashed arrows). Each mask is assigned a semantic class, resulting in a pseudo label. The examples are taken from the Cityscapes *(top)*, COCO-Stuff *(middle)*, and Potsdam-3 *(bottom)* datasets.

Despite the challenges of unsupervised semantic segmentation, we have witnessed remarkable progress on this task in the past years (Ji et al., 2019; Cho et al., 2021; Van Gansbeke et al., 2021; 2022; Ke et al., 2022; Yin et al., 2022; Hamilton et al., 2022; Karlsson et al., 2022; Li et al., 2023; Seong et al., 2023; Seitzer et al., 2023). Deep representations obtained with self-supervised learning (SSL), such as DINO (Caron et al., 2021), have played a critical role in this advance. However, it remains unclear whether previous work leverages the intrinsic properties of the original SSL representations, or merely uses them for "bootstrapping" and learns a new representation on top. Exploiting the inherent properties of SSL features is preferable for two reasons. First, training SSL models incurs a substantial computational effort, justifiable only if the learned feature extractor is sufficiently versatile. In other words, one can amortize the high computational cost over many downstream tasks, provided that task specialization is computationally negligible. Second, studying SSL representations with lightweight tools, such as linear models, leads to a more interpretable empirical analysis than with the use of more complex models, as evidenced by the widespread use of linear probing in SSL evaluation. Such interpretability advances research on SSL models toward improved cross-task generalization.

Equipped with essential tools of linear modeling, *i. e.* Principal Component Analysis (PCA), we generate **Pri**ncipal **Ma**sk **P**roposal**s**, or PriMaPs, directly from the SSL representation. Complementing previous findings on object-centric images (Tumanyan et al., 2022; Amir et al., 2022), we show that principal components of SSL features tend to identify visual patterns with high semantic correlation also in scene-centric imagery. Leveraging PriMaPs and minimalist post-processing, we construct semantic pseudo labels for each image as illustrated in Fig. 1. Finally, instead of learning a new embedding on top of the SSL representation (Hamilton et al., 2022; Seong et al., 2023; Seitzer et al., 2023; Zadaianchuk et al., 2023), we employ a moving average implementation of stochastic Expectation Maximization (EM) (Chen et al., 2018a) to assign a consistent category to each segment in the pseudo labels and directly optimize class prototypes in the feature space. Our experiments show that this straightforward approach not only boosts the segmentation accuracy of the DINO baseline, but also that of more advanced state-of-the-art approaches tailored for semantic segmentation, such as STEGO (Hamilton et al., 2022) and HP (Seong et al., 2023).

We make the following contributions: *(i)* We derive lightweight mask proposals, leveraging intrinsic properties of the embedding space, *e. g.*, the covariance matrix, provided by an off-the-shelf SSL approach. *(ii)* Based on the mask proposals, we construct pseudo labels and employ moving average stochastic EM to assign a consistent semantic class to each proposal. *(iii)* We demonstrate improved segmentation accuracy across a wide range of SSL embeddings and datasets.

## 2 Related Work

Our work builds upon recent advances in self-supervised representation learning, and takes inspiration from previous unsupervised semantic and instance segmentation methods.

The goal of **self-supervised representation learning (SSL)** is to provide generic, task-agnostic feature extractors (He et al., 2020; Chen et al., 2020; Grill et al., 2020). A pivotal role in defining the behavior of self-supervised features on future downstream tasks is taken by the self-supervised objective, the so-called *pretext* task. Examples of such tasks include predicting the context of a patch (Doersch et al., 2015) or its rotation (Gidaris et al., 2018), image inpainting (Pathak et al., 2016), and "solving" jigsaw puzzles (Noroozi & Favaro, 2016). Another family of self-supervised techniques is based on contrastive learning (Chen et al., 2020; Caron et al., 2020). More recently, Transformer networks (Dosovitskiy et al., 2020) revived some older pretext tasks, such as context prediction (Caron et al., 2021; He et al., 2022), in a more data-scalable fashion.

While the standard evaluation practice in SSL (*e. g.*, linear probing, transfer learning) offers some glimpse into the features' properties, understanding the embedding space produced by SSL remains an active terrain for research (Ericsson et al., 2021; Naseer et al., 2021). In particular, DINO features (Caron et al., 2021; Oquab et al., 2024) are known to encode accurate object-specific information, such as object parts (Amir et al., 2022; Tumanyan et al., 2022). However, it remains unclear to what extent DINO embeddings allow for semantic representation of the more ubiquitous multi-object scenes. Here, following previous work (*e. g.*, Hamilton et al., 2022; Seong et al., 2023), we provide further insights.

Early techniques for **unsupervised semantic segmentation** using deep networks (Cho et al., 2021; Van Gansbeke et al., 2021) approach the problem in the spirit of transfer learning and, under certain nomenclatures, may not be considered fully unsupervised. Specifically, starting with *supervised* ImageNet pre-training (Russakovsky et al., 2015), a network obtains a fine-tuning signal from segmentation-oriented training objectives. Such supervised "bootstrapping" appears to be crucial in the ill-posed unsupervised formulation. Unsupervised training of a deep model for segmentation from scratch is possible, albeit sacrificing accuracy (Ji et al., 2019; Ke et al., 2022). However, training a new deep model for each downstream task contradicts the spirit of SSL of amortizing the high SSL training costs over many computationally cheap specializations of the learned features (Bommasani et al., 2021).

Relying on *self-supervised* DINO pre-training, recent work (Hamilton et al., 2022; Li et al., 2023; Seong et al., 2023) has demonstrated the potential of such amortization with more lightweight fine-tuning for semantic segmentation. Nevertheless, most of this work has treated the SSL representation as an inductive prior by learning a new embedding space over the SSL features (*e. g.*, Hamilton et al., 2022; Seong et al., 2023). In contrast, following SSL principles, we use the SSL representation in a more direct and lightweight fashion – by extracting mask proposals using linear models (PCA) with minimal post-processing and learning a direct mapping from feature to prediction space.

Mask proposals have an established role in computer vision (Arbelaez et al., 2011; Uijlings et al., 2013), and remain highly relevant in deep learning (Hwang et al., 2019; Van Gansbeke et al., 2021; Yin et al., 2022). Different from previous work, we directly derive the mask proposals from SSL representations. Our approach is inspired by the recent use of classical algorithms, such as normalized cuts (Ncut Shi & Malik, 2000), in the context of self-supervised segmentation (Wang et al., 2023a;b). However previous approaches (Van Gansbeke et al., 2021; 2022; Wang et al., 2023a;b) mainly proposed foreground object masks on object-centric data, utilized in a multi-step self-training. In contrast, we develop a straightforward method for extracting dense pseudo labels for learning unsupervised semantic segmentation of scene-centric data and show consistent benefits in improving the segmentation accuracy across a variety of baselines and state-of-the-art methods (Hamilton et al., 2022; Seong et al., 2023).

## 3 PriMaPs: Principal Mask Proposals

In this paper, we leverage recent advances in self-supervised representation learning (Caron et al., 2021; Oquab et al., 2024) for the specific downstream task of unsupervised semantic segmentation. Our approach is

based on the observation that such pre-trained features already exhibit intrinsic spatial similarities, capturing semantic correlations, thus providing guidance to fit global pseudo-class representations.

**A simple baseline.** Consider a simple baseline that applies $K$-means clustering to DINO ViT features (Caron et al., 2021). Surprisingly, this already leads to reasonably good unsupervised semantic segmentation results, *e.g.*, around 15 % mean IoU to segment 27 classes on Cityscapes (Cordts et al., 2016), see Tab. 1. However, supervised linear probing between the same feature space and the ground-truth labels – the theoretical upper bound – leads to clearly superior results of almost 36 % mean IoU. Given this gap and the simplicity of the approach, we conclude that there is *valuable potential* in directly obtaining semantic segmentations without enhancing the original feature representation, unlike in previous work (Hamilton et al., 2022; Seong et al., 2023).

**From $K$-means to PriMaPs-EM.** When examining the $K$-means baseline as well as state-of-the-art methods (Hamilton et al., 2022; Seong et al., 2023), see Fig. 4, it can be qualitatively observed that more local consistency within the respective predictions would already lead to less mis-classification. We take inspiration from (Drineas et al., 2004; Ding & He, 2004), who showed that the PCA subspace, spanned by principal components, is a relaxed solution to $K$-means clustering. We observe that principal components have high semantic correlation for object- as well as scene-centric image features (*cf.* Fig. 1). We utilize this by iteratively partitioning images based on dominant feature patterns, identified by means of the cosine similarity of the image features to the respective first principal component. We name the resulting class-agnostic image decomposition *PriMaPs* – Principal Mask Proposals. PriMaPs stem directly from SSL representations and guide the process of unsupervised semantic segmentation. Shown in Fig. 3, our optimization-based approach, PriMaPs-EM, operates over an SSL feature representation computed from a frozen deep neural network backbone. The optimization realizes stochastic EM of a clustering objective guided by PriMaPs. Specifically, PriMaPs-EM fits class prototypes to the proposals in a globally consistent manner by optimizing over two identically sized vector sets, with one of them being an exponential moving average (EMA) of the other. We show that PriMaPs-EM enables accurate unsupervised partitioning of images into semantically meaningful regions while being comparatively lightweight and orthogonal to most previous approaches to unsupervised semantic segmentation.

## 3.1 Deriving PriMaPs

We start with a frozen pre-trained self-supervised backbone model $\mathcal{F} : \mathbb{R}^{3 \times h \times w} \to \mathbb{R}^{C \times H \times W}$, which embeds an image $I \in \mathbb{R}^{3 \times h \times w}$ into a dense feature representation $f \in \mathbb{R}^{C \times H \times W}$ as

$$f = \mathcal{F}(I) \,. \tag{1}$$

Here, $C$ refers to the channel dimension of the dense features, and $H = {}^h\!/_p, W = {}^w\!/_p$ with $p$ corresponding to the output stride of the backbone. Based on this image representation, the next step is to decompose the image into semantically meaningful masks to provide a local grouping prior for fitting global class prototypes.

**Initial principal mask proposal.** To identify the initial principal mask proposal in an image $I$, we analyze the spatial statistical correlations of its features by means of PCA. Specifically, we consider the empirical feature covariance matrix

$$\Sigma = \frac{1}{HW} \sum_{i=1}^{H} \sum_{j=1}^{W} \big(f_{:,i,j} - \bar{f}\big)\big(f_{:,i,j} - \bar{f}\big)^{\top} \,, \tag{2}$$

where $f_{:,i,j} \in \mathbb{R}^C$ are the features at position $(i,j)$ and $\bar{f} \in \mathbb{R}^C$ is the mean feature. To identify the feature direction that captures the largest variance in the feature distribution, we seek the first principal component of $\Sigma$ by solving

$$\Sigma v = \lambda v \,. \tag{3}$$

We obtain the first principal component as the eigenvector $v_1$ to the largest eigenvalue $\lambda_1$, which can be computed efficiently with Singular Value Decomposition (SVD) using the flattened features $f$.

To identify a candidate region, our next goal is to compute a spatial feature similarity map to the dominant feature direction. We observe that doing so directly with the principal direction does not always lead

Figure 2: **PriMaPs process.** Given the dense feature embeddings $f$ of an image $I$, we compute the cosine-similarity map $M^1$ of all features $f$ to their first principal component's nearest neighbor feature. The first PriMaP $P^1$ is obtained by thresholding $M^1$. To obtain $P^2$, the features assigned to $P^1$ are masked out, and the process is repeated with the remaining features $f^2$. We repeat the PriMaPs process until the majority of features have been assigned to masks. Finally, all masks $P$ are upsampled and refined using a CRF.

to sufficiently good localization, *i. e.*, high similarities arise across multiple visual concepts in an image, elaborated in more detail in Appendix A.1. This can be circumvented by first anchoring the dominant feature vector in the feature map. To that end, we obtain the nearest neighbor feature $\tilde{f} \in \mathbb{R}^C$ of the first principal component $v_1$ by considering the cosine distance in the normalized feature space $\hat{f}$ as

$$\tilde{f} = \hat{f}_{:,m,n}, \quad \text{where} \quad (m,n) = \arg\max_{i,j} \left(v_1^\top \hat{f}\right). \tag{4}$$

Given this, we compute the cosine-similarity map $M^1 \in \mathbb{R}^{H \times W}$ of the dominant feature *w. r. t.* all features as

$$M^1 = (M_{i,j})_{i,j}, \quad \text{where} \quad M_{i,j} = \left(\tilde{f}\right)^\top \hat{f}_{:,i,j}. \tag{5}$$

Next, a threshold $\psi \in (0,1)$ is applied to the similarity map in order to suppress noise and further localize the initial mask. Accordingly, elements of a binary similarity map $P^1 \in \{0,1\}^{H \times W}$ are set to 1 when the similarity is larger than a fraction $\psi$ of the maximal similarity, and 0 otherwise, *i. e.*,

$$P^1 = \left[M_{i,j}^1 > \psi \cdot \max_{m,n} M_{m,n}^1\right]_{i,j}, \tag{6}$$

where $[\cdot]$ denotes the Iverson bracket. This binary *principal mask* $P^1$ gives rise to the first principal mask proposal in image $I$.

**Further principal mask proposals.** Subsequent mask proposals result from iteratively repeating the described procedure. To that end, it is necessary to suppress features that have already been assigned to a pseudo label. Specifically in iteration $z$, given the mask proposals $P^s$, $s = 1, \ldots, z-1$, extracted in previous iterations, we mask out the features that have already been considered as

$$f_{:,i,j}^z = f_{:,i,j} \left[\sum_{s=1}^{z-1} P_{i,j}^s = 0\right]. \tag{7}$$

Applying Eqs. (2) to (6) on top of the masked features $f^z$ yields the cosine-similarity map $M^z$ and the principal mask proposal $P^z$, and so on. We repeat this procedure until the majority of features (*e. g.*, 95%) have been assigned to a mask. In a final step, the remaining features, in case there are any, are assigned to an "ignore" mask

$$P_{i,j}^0 = 1 - \sum_{z=1}^{Z-1} P_{i,j}^z. \tag{8}$$

This produces a tensor $P \in \{0,1\}^{Z \times H \times W}$ of $Z$ spatial similarity masks, decomposing a single image into $Z$ non-overlapping regions.

**Proposal post-processing.** To further improve the alignment of the masks with edges and color-correlated regions in the image, a fully connected Conditional Random Field (CRF) with Gaussian edge potentials (Krähenbühl & Koltun, 2011) is applied to the initial mask proposals $P$ (after bilinear upsampling to the image resolution) for 10 inference iterations. The process for obtaining PriMaPs is visualized in Fig. 2.

**PriMaPs pseudo-label generation.** In order to form a pseudo label for semantic segmentation out of the $Z$ class-agnostic mask proposals, each mask has to be assigned one out of $K$ pseudo-class labels. This

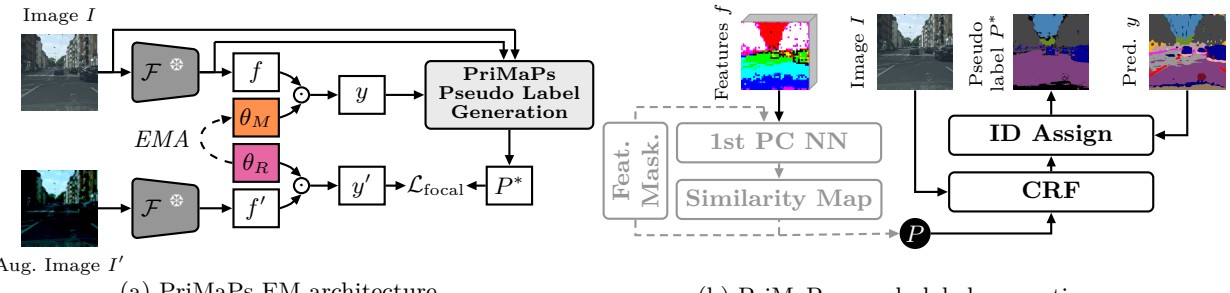

(a) PriMaPs-EM architecture

(b) PriMaPs pseudo-label generation

Figure 3: **(a) PriMaPs-EM architecture.** An image $I$ and its augmented version $I'$ are embedded by the frozen self-supervised backbone $\mathcal{F}$, resulting in the dense features $f$ and $f'$. The segmentation prediction $y$ by the momentum class prototypes $\theta_M$ arises via the dot product with $f$. Likewise, $y'$ arises from the dot product of the running class prototypes $\theta_R$ with $f'$. Pseudo labels $P^*$ are constructed using PriMaPs, $I$, and $y$. We use the pseudo labels to optimize $\theta_R$, applying a focal loss. $\theta_R$ is gradually transferred to $\theta_M$ by means of an EMA. **(b) PriMaPs pseudo-label generation.** Masks $P$ are proposed by iterative binary partitioning based on the cosine similarity of the features of any unassigned pixel to their first principal component's nearest neighbor feature. Gray indicates these iterative steps. Next, the masks $P$ are aligned to the image $I$ using a CRF. Finally, a per-mask pseudo-class ID is assigned using majority voting based on the segmentation prediction $y$, resulting in the pseudo label $P^*$.

is accomplished using a segmentation prediction of our optimization process, called PriMaPs-EM, detailed in Sec. 3.2. Given a segmentation prediction $y$, we assign the pseudo-class ID that is most frequently predicted within each proposal, yielding the final pseudo-label map $P^* \in \{0,1\}^{K \times h \times w}$, a one-hot encoding of a pseudo-class ID. The entire PriMaPs pseudo-label generation process is illustrated in Figure 3b.

## 3.2 PriMaPs-EM

Shown in Fig. 3, PriMaPs-EM is an iterative optimization technique for fitting class prototypes for semantic segmentation across a dataset utilizing pseudo labels $P^*$ based on PriMaPs (*cf.* Sec. 3.1). PriMaPs provide guidance through local per-image consistency for fitting the global class prototypes across a dataset. PriMaPs-EM leverages a frozen pre-trained self-supervised backbone model $\mathcal{F}$ and optimizes two identically sized vector sets, the running class prototypes $\theta_R$ and their moving average, the momentum class prototypes $\theta_M$. The class prototypes $\theta_R$ and $\theta_M$ are the $K$ pseudo-class representations in the feature space, projecting the $C$-dimensional features linearly to $K$ semantic pseudo classes, which equates to segmenting. More precisely, we use the segmentation prediction of the momentum class prototypes $\theta_M$ to assign consistent semantic pseudo-class IDs to the class-agnostic PriMaPs. We optimize the running class prototypes $\theta_R$ using the pseudo labels $P^*$ and gradually transfer $\theta_R$ to $\theta_M$ using an EMA.

PriMaPs-EM performs the optimization in two stages, since in our case, a meaningful initialization of the class prototypes is vital to provide a reasonable optimization signal. This can be traced back to the pseudo-label generation, which utilizes a segmentation prediction to assign globally consistent classes to the masks. Initializing the class prototypes randomly leads to a highly unstable and noisy signal.

**Initialization.** We initialize the class prototypes $\theta_M$ with the first $K$ principal components using vanilla PCA computed over the feature embeddings of a large number of images across the respective dataset. Next, a cosine-distance batch-wise $K$-means (MacQueen, 1967) loss

$$\mathcal{L}_{K\text{-means}}(\theta_M) = -\sum_{i,j} \max\left(\theta_M^\top f_{:,i,j}\right) \tag{9}$$

is minimized with respect to $\theta_M$ for a fixed number of epochs. This minimizes the cumulative cosine distances of the image features $f_{:,i,j}$ to their respective closest class prototype. $\theta_R$ is initialized with the same prototypes.

**Moving average stochastic EM.** Visualized in Fig. 3a, every optimization iteration starts with computing the segmentation prediction $y$ by the dot product of the dense image features $f$ of the image $I$ with the momentum class prototypes $\theta_M$. Applying the $\arg\max$ results in the dominant prototype for each feature location and, accordingly, in prototype-based binary masks $y \in \{0,1\}^{K \times h \times w}$. Note that we bilinearly upsample the features to the image resolution before the dot product. Using $y$, $f$, and $I$, we get PriMaPs pseudo labels $P^*$ as described in Sec. 3.1. We obtain the semantic prediction $y'$ by the dot product of the running class prototypes $\theta_R$ with the dense image features $f'$ of the augmented image $I'$. $\theta_R$ is optimized by applying a batch-wise focal loss (Lin et al., 2020) with respect to the pseudo labels $P^*$. The focal loss $\mathcal{L}_{\text{focal}}$ is a weighted version of the cross-entropy loss, increasing the loss contribution of less confident classes, *i.e.*,

$$\mathcal{L}_{\text{focal}}(\theta_R; y') = -\sum_{k,i,j}(1 - \chi_k)^2 P^*_{k,i,j} \log(y'_{k,i,j}), \tag{10}$$

where $y'_{k,i,j} = \mathrm{softmax}(\theta_R^\top f'_{:,i,j})$ are the predictions and $\chi_k$ is the class-wise confidence value approximated by averaging $y'_{k,:,:}$ spatially. The running class prototypes $\theta_R$ are optimized with an augmented input image $I'$. We employ photometric augmentations (Gaussian blur, grayscaling, and color jitter), introducing a controlled noise, thereby strengthening the robustness of our class representation. The momentum class prototypes $\theta_M$ are the EMA of the running class prototypes $\theta_R$. This is utilized in order to stabilize the optimization, accounting for the noisy nature of the unsupervised signal used for optimization. We update $\theta_M$ every $\kappa$ iterations with a decay $\gamma$ as

$$\theta_M^{t+\kappa} = \gamma \theta_M^t + (1 - \gamma)\theta_R^{t+\kappa}, \tag{11}$$

where $t$ is the iteration index of the previous update. This optimization approach resembles moving average stochastic EM. Hereby, the E-step amounts to finding pseudo labels using PriMaPs and the momentum class prototypes. The M-step optimizes the running class prototypes with respect to their focal loss $\mathcal{L}_{\text{focal}}$. Stochasticity arises from performing EM in mini-batches of images.

**Inference.** At inference time, we obtain the segmentation prediction $y$ via the momentum class prototypes $\theta_M$, refined using a fully connected CRF with Gaussian edge potentials (Krähenbühl & Koltun, 2011) following previous approaches (Van Gansbeke et al., 2021; Hamilton et al., 2022; Seong et al., 2023). This is the identical CRF as already used for refining the masks in the PriMaPs pseudo-label generation using the same CRF parameters as previous work (Van Gansbeke et al., 2021; Hamilton et al., 2022; Seong et al., 2023).

## 4  Experiments

To assess the efficacy of our approach, we compare it to the current state-of-the-art in unsupervised semantic segmentation. For a fair comparison, we closely follow the overall setup used by numerous previous works (Ji et al., 2019; Cho et al., 2021; Hamilton et al., 2022; Seong et al., 2023).

### 4.1  Experimental Setup

**Datasets.** Following the practice of previous work, we conduct experiments on Cityscapes (Cordts et al., 2016), COCO-Stuff (Caesar et al., 2018), and Potsdam-3 (ISPRS). Cityscapes and COCO-Stuff are evaluated using 27 classes, while Potsdam is evaluated on the 3-class variant. Adopting the established evaluation protocol (Ji et al., 2019; Cho et al., 2021; Hamilton et al., 2022; Seong et al., 2023), we resize images to 320 pixels along the smaller axis and crop the center $320 \times 320$ pixels. This is adjusted to 322 pixels for DINOv2. Different from previous work, we apply this simple scheme throughout this work, thus dispensing with elaborate multi-crop approaches of previous methods (Hamilton et al., 2022; Yin et al., 2022; Seong et al., 2023).

**Self-supervised backbone.** Experiments are conducted across a collection of pre-trained self-supervised feature embeddings: DINO (Caron et al., 2021) based on ViT-Small and ViT-Base using $8 \times 8$ patches; and DINOv2 (Oquab et al., 2024) based on ViT-Small and ViT-Base using $14 \times 14$ patches. In the spirit of SSL principles, we keep the backbone parameters frozen throughout the experiments. We use the output from the last network layer as our SSL feature embeddings. Since PriMaPs-EM is agnostic to the used embedding

Table 1: **Cityscapes – PriMaPs-EM *(Ours)* comparison to existing unsupervised semantic segmentation methods**, using Accuracy and mean IoU (in %) for unsupervised and supervised probing. Double citations refer to a method's origin and the work conducting the experiment.

| Method | Backbone | Unsupervised | | Supervised | |
| --- | --- | --- | --- | --- | --- |
| | | **Acc** | **mIoU** | Acc | mIoU |
| IIC (Ji et al., 2019; Cho et al., 2021) | ResNet18 +FPN | 47.9 | 6.4 | – | – |
| MDC (Caron et al., 2018; Cho et al., 2021) | | 40.7 | 7.1 | – | – |
| PiCIE (Cho et al., 2021) | | 65.5 | 12.3 | – | – |
| VICE (Karlsson et al., 2022) | | 31.9 | 12.8 | 86.3 | 31.6 |
| Baseline (Caron et al., 2021) | DINO ViT-S/8 | 61.4 | 15.8 | 91.0 | 35.4 |
| + TransFGU (Yin et al., 2022) | | 77.9 | 16.8 | – | – |
| + HP (Seong et al., 2023) | | 80.1 | 18.4 | 91.2 | 30.6 |
| + PriMaPs-EM | | **81.2** | **19.4** | 91.0 | 35.4 |
| + HP (Seong et al., 2023) + PriMaPs-EM | | 76.6 | 19.2 | 91.2 | 30.6 |
| Baseline (Caron et al., 2021) | DINO ViT-B/8 | 49.2 | 15.5 | 91.6 | 35.9 |
| + STEGO (Hamilton et al., 2022; Koenig et al., 2023) | | 73.2 | 21.0 | 89.6 | 28.0 |
| + HP (Seong et al., 2023) | | **79.5** | 18.4 | 90.9 | 33.0 |
| + PriMaPs-EM | | 59.6 | 17.6 | 91.6 | 35.9 |
| + STEGO (Hamilton et al., 2022) + PriMaPs-EM | | 78.6 | **21.6** | 89.6 | 28.0 |
| Baseline (Oquab et al., 2024) | DINOv2 ViT-S/14 | 49.5 | 15.3 | 90.8 | 41.9 |
| + PriMaPs-EM | | **71.5** | **19.0** | 90.8 | 41.9 |
| Baseline (Oquab et al., 2024) | DINOv2 ViT-B/14 | 36.1 | 14.9 | 91.0 | 44.8 |
| + PriMaPs-EM | | **82.9** | **21.3** | 91.0 | 44.8 |

space, we can also apply it on top of current state-of-the-art unsupervised segmentation pipelines. Here, we consider STEGO (Hamilton et al., 2022) and HP (Seong et al., 2023), which also use DINO features but learn a target domain-specific subspace.

**Baseline.** Following (Hamilton et al., 2022; Seong et al., 2023), we train a single linear layer as a baseline with the same structure as $\theta_R$ and $\theta_M$ by minimizing the cosine-distance batch-wise $K$-Means loss from Eq. (9). Hereby, parameters, such as the number of epochs and the learning rate, are identical to those used when employing PriMaPs-EM.

**PriMaPs-EM.** As discussed in Sec. 3.2, the momentum class prototypes $\theta_M$ are initialized using the first $K$ principal components; we use 2975 images for PCA, as this is the largest number of training images shared by all datasets. Next, $\theta_M$ is pre-trained by minimizing Eq. (9) using Adam (Kingma & Ba, 2015). We use a learning rate of 0.005 for 2 epochs on all datasets and backbones. The weights are then copied to $\theta_R$. For fitting the running class prototypes using EM, $\theta_R$ is optimized by minimizing the focal loss from Eq. (10) with Adam (Kingma & Ba, 2015) using a learning rate of 0.005. The momentum class prototypes $\theta_M$ are updated using an EMA according to Eq. (11) every $\gamma_s = 10$ steps with decay $\gamma_\psi = 0.98$. We set the PriMaPs mask-proposal threshold to $\psi = 0.4$ and provide detailed ablation experiments in Appendix A.2. We use a batch size of 32 for 50 epochs on Cityscapes and Potsdam-3, and use 5 epochs on COCO-Stuff due to its larger size. Importantly, the same hyperparameters are used across *all* datasets and backbones. Moreover, note that fitting class prototypes with PriMaPs-EM is quite practical, *e.g.*, about 2 hours on Cityscapes. Experiments are conducted on a single NVIDIA A6000 GPU.

**Supervised upper bounds.** To assess the potential of the SSL features used, we report supervised upper bounds. Specifically, we train a linear layer using cross entropy and Adam with a learning rate of 0.005. Since PriMaPs-EM uses frozen SSL features, its supervised bound is the same as that of the underlying features. This is not the case, however, for prior work (Hamilton et al., 2022; Seong et al., 2023), which project the feature representation affecting the upper bound.

**Evaluation.** For inference, we use the prediction from the momentum class prototypes $\theta_M$. CRF refinement uses 10 inference iterations and standard parameters $a = 4, b = 3, \theta_\alpha = 67, \theta_\beta = 3, \theta_\gamma = 1$ from prior work

Table 2: **COCO-Stuff – PriMaPs-EM *(Ours)* comparison to existing unsupervised semantic segmentation methods**, using Accuracy and mean IoU (in %) for unsupervised and supervised probing. Double citations refer to a method's origin and the work conducting the experiment.

| Method | Backbone | Unsupervised | | Supervised | |
|---|---|---|---|---|---|
| | | Acc | mIoU | Acc | mIoU |
| IIC (Ji et al., 2019; Cho et al., 2021) | ResNet18 +FPN | 21.8 | 6.7 | 44.5 | 8.4 |
| MDC (Caron et al., 2018; Cho et al., 2021) | | 32.2 | 9.8 | 48.6 | 13.3 |
| PiCIE (Cho et al., 2021) | | 48.1 | 13.8 | 54.2 | 13.9 |
| PiCIE+H (Cho et al., 2021) | | 50.0 | 14.4 | 54.8 | 14.8 |
| VICE (Karlsson et al., 2022) | | 28.9 | 11.4 | 62.8 | 25.5 |
| Baseline (Caron et al., 2021) | DINO ViT-S/8 | 34.2 | 9.5 | 72.0 | 41.3 |
| + TransFGU (Yin et al., 2022) | | 52.7 | 17.5 | – | – |
| + STEGO (Hamilton et al., 2022) | | 48.3 | 24.5 | 74.4 | 38.3 |
| + ACSeg (Li et al., 2023) | | – | 16.4 | – | – |
| + HP (Seong et al., 2023) | | 57.2 | 24.6 | 75.6 | 42.7 |
| + PriMaPs-EM | | 46.5 | 16.4 | 72.0 | 41.3 |
| + HP (Seong et al., 2023) + PriMaPs-EM | | **57.8** | **25.1** | 75.6 | 42.7 |
| Baseline (Caron et al., 2021) | DINO ViT-B/8 | 38.8 | 15.7 | 74.0 | 44.6 |
| + STEGO (Hamilton et al., 2022) | | 56.9 | 28.2 | 76.1 | 41.0 |
| + PriMaPs-EM | | 48.5 | 21.9 | 74.0 | 44.6 |
| + STEGO (Hamilton et al., 2022) + PriMaPs-EM | | **57.9** | **29.7** | 76.1 | 41.0 |
| Baseline (Oquab et al., 2024) | DINOv2 ViT-S/14 | 44.5 | 22.9 | 77.9 | 52.8 |
| + PriMaPs-EM | | **46.5** | **23.8** | 77.9 | 52.8 |
| Baseline (Oquab et al., 2024) | DINOv2 ViT-B/14 | 35.0 | 17.9 | 77.3 | 53.7 |
| + PriMaPs-EM | | **52.8** | **23.6** | 77.3 | 53.7 |

(Van Gansbeke et al., 2021; Hamilton et al., 2022; Seong et al., 2023). We evaluate common metrics in unsupervised semantic segmentation, specifically the mean Intersection over Union (mIoU) and Accuracy (Acc) over all classes after aligning the predicted class IDs with ground-truth labels by means of Hungarian matching (Kuhn, 1955).

**SotA + PriMaPs-EM.** To explore our method's potential, we additionally employ PriMaPs-EM on top of STEGO (Hamilton et al., 2022) and HP (Seong et al., 2023). For each backbone-dataset combination, we apply it on top of the best previous method in terms of mIoU. To that end, the training signal for learning the feature projection of (Hamilton et al., 2022; Seong et al., 2023) remains unchanged. We apply PriMaPs-EM fully orthogonally, using the DINO backbone features for pseudo-label generation and fit a direct connection between the feature space of the state-of-the-art method and the prediction space.

### 4.2 Results

We compare PriMaPs-EM against prior work for unsupervised semantic segmentation (Ji et al., 2019; Cho et al., 2021; Hamilton et al., 2022; Yin et al., 2022; Li et al., 2023; Seong et al., 2023). As in previous work, we use DINO (Caron et al., 2021) as the main baseline. Additionally, we also test PriMaPs-EM on top of DINOv2 (Oquab et al., 2024), STEGO (Hamilton et al., 2022), and HP (Seong et al., 2023). Overall, we observe that the DINO baseline already achieves strong results (*cf.* Tabs. 1 to 3). DINOv2 features significantly raise the supervised upper bounds in terms of Acc and mIoU, the improvement in the unsupervised case remains more modest. Nevertheless, PriMaPs-EM further boosts the unsupervised segmentation performance.

In Tab. 1, we compare to previous work on the Cityscapes dataset. PriMaPs-EM leads to a consistent improvement over all baselines in terms of unsupervised segmentation accuracy. For example, PriMaPs-EM boosts DINO ViT-S/8 by +3.6% and +19.8% in terms of mIoU and Acc, respectively, which leads to state-of-the-art performance. Notably, we find PriMaPs-EM to be complementary to other state-of-the-art unsupervised segmentation methods like STEGO (Hamilton et al., 2022) and HP (Seong et al., 2023) on the

Table 3: **Potsdam-3 − PriMaPs-EM *(Ours)* comparison to existing unsupervised semantic segmentation methods**, using Accuracy and mean IoU (in %) for unsupervised and supervised probing. Double citations refer to a method's origin and the work conducting the experiment.

| Method | Backbone | Unsupervised | | Supervised | |
|---|---|---|---|---|---|
| | | Acc | mIoU | Acc | mIoU |
| RandomCNN (Cho et al., 2021) | VGG 11 | 38.2 | – | – | – |
| K-Means (Pedregosa et al., 2011; Cho et al., 2021) | | 45.7 | – | – | – |
| SIFT (Lowe, 2004; Cho et al., 2021) | | 38.2 | – | – | – |
| ContextPrediction (Doersch et al., 2015; Cho et al., 2021) | | 49.6 | – | – | – |
| CC (Isola et al., 2015; Cho et al., 2021) | | 63.9 | – | – | – |
| DeepCluster (Caron et al., 2018; Cho et al., 2021) | | 41.7 | – | – | – |
| IIC (Ji et al., 2019; Cho et al., 2021) | | 65.1 | – | – | – |
| Baseline (Caron et al., 2021) | DINO ViT-S/8 | 56.6 | 33.6 | 82.0 | 69.0 |
| +STEGO (Hamilton et al., 2022; Koenig et al., 2023) | | 77.0 | 62.6 | 85.9 | 74.8 |
| +PriMaPs-EM | | 62.5 | 38.9 | 82.0 | 69.0 |
| +STEGO (Hamilton et al., 2022)+PriMaPs-EM | | **78.4** | **64.2** | 85.9 | 74.8 |
| Baseline (Caron et al., 2021) | DINO ViT-B/8 | 66.1 | 49.4 | 84.3 | 72.8 |
| +HP (Seong et al., 2023) | | 82.4 | 69.1 | 88.0 | 78.4 |
| +PriMaPs-EM | | 80.5 | 67.0 | 84.3 | 72.8 |
| +HP (Seong et al., 2023)+PriMaPs-EM | | **83.3** | **71.0** | 88.0 | 78.4 |
| Baseline (Oquab et al., 2024) | DINOv2 ViT-S/14 | 75.9 | 61.0 | 86.6 | 76.2 |
| +PriMaPs-EM | | **78.5** | **64.3** | 86.6 | 76.2 |
| Baseline (Oquab et al., 2024) | DINOv2 ViT-B/14 | 82.4 | 69.9 | 87.9 | 78.3 |
| +PriMaPs-EM | | **83.2** | **71.1** | 87.9 | 78.3 |

corresponding backbone model. This suggests that these methods use their SSL representation only to a limited extent and do not fully leverage the inherent properties of the underlying SSL embeddings. Similar observations can be drawn for the experiments on COCO-Stuff in Tab. 2. PriMaPs-EM leads to a consistent improvement across all four SSL baselines, as well as an improvement over STEGO and HP. For instance, combining STEGO with PriMaPs-EM leads to +14.0% and +19.1% improvement over the baseline in terms of mIoU and Acc for DINO ViT-B/8. Experiments on the Potsdam-3 dataset follow the same pattern (*cf.* Tab. 3). PriMaPs-EM leads to a consistent gain over the baseline, *e. g.* +17.6% and +14.4% in terms of mIoU and Acc, respectively, for DINO ViT-B/8. Moreover, it also boosts the accuracy of STEGO and HP. In some cases, the gain of PriMaPs-EM is limited. For example, in Tab. 1 for DINO ViT-B/8 + PriMaPs-EM, the class prototype for "sidewalk" is poor while the classes "road" and "vegetation" superimpose smaller objects. For DINO ViT-S/8 + PriMaPs-EM in Tab. 3, the class prototype "road" is poor. This limits the overall performance of our method while still outperforming the respective baseline in both cases.

Overall, PriMaPs-EM provides modest but consistent benefits over a wide range of baselines and datasets and reaches competitive segmentation performance *w. r. t.* the state-of-the-art using identical hyperparameters across *all* backbones and datasets. Recalling the simplicity of the techniques behind PriMaPs, we believe that this is a significant result. The complementary effect of PriMaPs-EM on other state-of-the-art methods (STEGO, HP) further suggests that they rely on DINO features for mere "bootstrapping" and learn feature representations with orthogonal properties to those of DINO. We conclude that PriMaPs-EM constitutes a straightforward, entirely orthogonal tool for boosting unsupervised semantic segmentation.

## 4.3 Ablation Study

To untangle the factors behind PriMaPs-EM, we examine the individual components in a variety of ablation experiments to access the contribution.

**PriMaPs pseudo-label ablations.** In Tab. 4a, we analyze the contribution of the individual sub-steps for PriMaPs pseudo-label generation by increasing the complexity of label generation. We provide the DINO

Table 4: **Ablation study** analyzing design choices and components in the PriMaPs pseudo-label generation *(a)* and PriMaPs-EM *(b)* for COCO-Stuff using DINO ViT-B/8.


<div>

(a) PriMaPs pseudo-label ablation

| Method | Acc | mIoU |
|---|---|---|
| Baseline (Caron et al., 2021) | 38.8 | 15.7 |
| Similarity Masks | 46.3 | 19.8 |
| + NN | 44.9 | 20.0 |
| + P-CRF ($\equiv$ PriMaPs-EM) | 48.4 | 21.9 |
| PriMaPs-EM(non-iter. PC) | 47.9 | 21.7 |

</div>
<div>

(b) PriMaPs-EM ablation

| Method | Acc | mIoU |
|---|---|---|
| Baseline (Caron et al., 2021) | 38.8 | 15.7 |
| + PriMaPs pseudo label | 38.8 | 18.0 |
| + EMA | 45.0 | 20.2 |
| + Augment | 46.0 | 20.4 |
| + CRF ($\equiv$ PriMaPs-EM) | 48.4 | 21.9 |

</div>
</div>

Table 5: **Oracle quality assessment of PriMaPs pseudo labels** for Cityscapes, COCO-Stuff, and Potsdam-3 by assigning oracle class IDs to the masks. "Pseudo" refers to evaluating only the pixels contained in the pseudo label, "All" to evaluating including the "ignore" assignments of the pseudo label.

| Method | *Cityscapes* | | *COCO-Stuff* | | *Potsdam-3* | |
|---|---|---|---|---|---|---|
| | **Acc** | **mIoU** | **Acc** | **mIoU** | **Acc** | **mIoU** |
| Pseudo | 92.4 | 54.0 | 93.4 | 82.4 | 95.2 | 90.9 |
| All | 73.2 | 32.4 | 74.1 | 55.9 | 67.4 | 48.9 |
| DINO ViT-B/8 Baseline (Caron et al., 2021) | 49.2 | 15.5 | 38.8 | 15.7 | 66.1 | 49.4 |

baseline, which corresponds to $K$-means feature clustering, for reference. In the most simplified case, we directly use the similarity mask, similar to Eq. (4). Next, we use the nearest neighbor (+NN in Tab. 4a) of the principal component to get the masks as in Eq. (5), followed by the full approach with CRF refinement (+P-CRF). Except for the changes in the pseudo-label generation, the optimization remains as described in Sec. 4.1. We observe that the similarity masks already provide a good staring point, yet we identify a gain from every single component step. This suggests that using the nearest neighbor improves the localization of the similarity mask. Similarly, CRF refinement improves the alignment between the masks and the image content. We also experiment with using the respective next principal component (non-iter. PC) instead of iteratively extracting the first principal component from the masked features. This leads to slightly inferior results. Naively using the K leading eigenvectors and simply assigning the masks based on the arg max of their cosine similarity to the features without any iterations would lead to significantly worse results with a pixel Accuracy of 43.1 % and a mIoU of 19.9 %. Note that nearest neighbor anchoring and thresholding are used in both experiments. Additionally, we ablate the nearest neighbor anchoring, the threshold $\psi$, and the stop criterion in Appendix A.

**PriMaPs-EM architecture ablations.** In a similar vein, we analyze the contribution of the different architectural components of PriMaPs-EM. Optimizing over a single set of class prototypes using the proposed PriMaPs pseudo labels already provides moderate improvement (+PriMaPs pseudo label in Tab. 4b), despite the disadvantage of an unstable and noisy optimization. Adding the EMA (+EMA) leads to a more stable optimization and further improved segmentation. Augmenting the input (+Augment) results in a further gradual improvement. We provide a more detailed breakdown of the augmentations in Appendix A.4. Similarly, refining the prediction with a CRF improves the results further (+CRF).

**Assessing PriMaPs pseudo labels.** To estimate the quality of the pseudo labels, respectively the principal masks, we decouple those from the class ID assignment by providing the oracle ground-truth class for each mask in Tab. 5. To that end, we evaluate all pixels included in our pseudo labels ("Pseudo"), corresponding to the upper bound of our optimization signal. Furthermore, we evaluate "All" by assigning the "ignore" pixels to a wrong class. The results indicate a high quality of the pseudo-label maps. We show qualitative examples of the PriMaPs mask proposals and pseudo labels in Appendix B.5.

**Qualitative results.** We show qualitative results for Cityscapes, COCO-Stuff, and Potsdam-3 in Fig. 4. We observe that PriMaPs-EM leads to less noisy results compared to the baseline, showcasing an improved

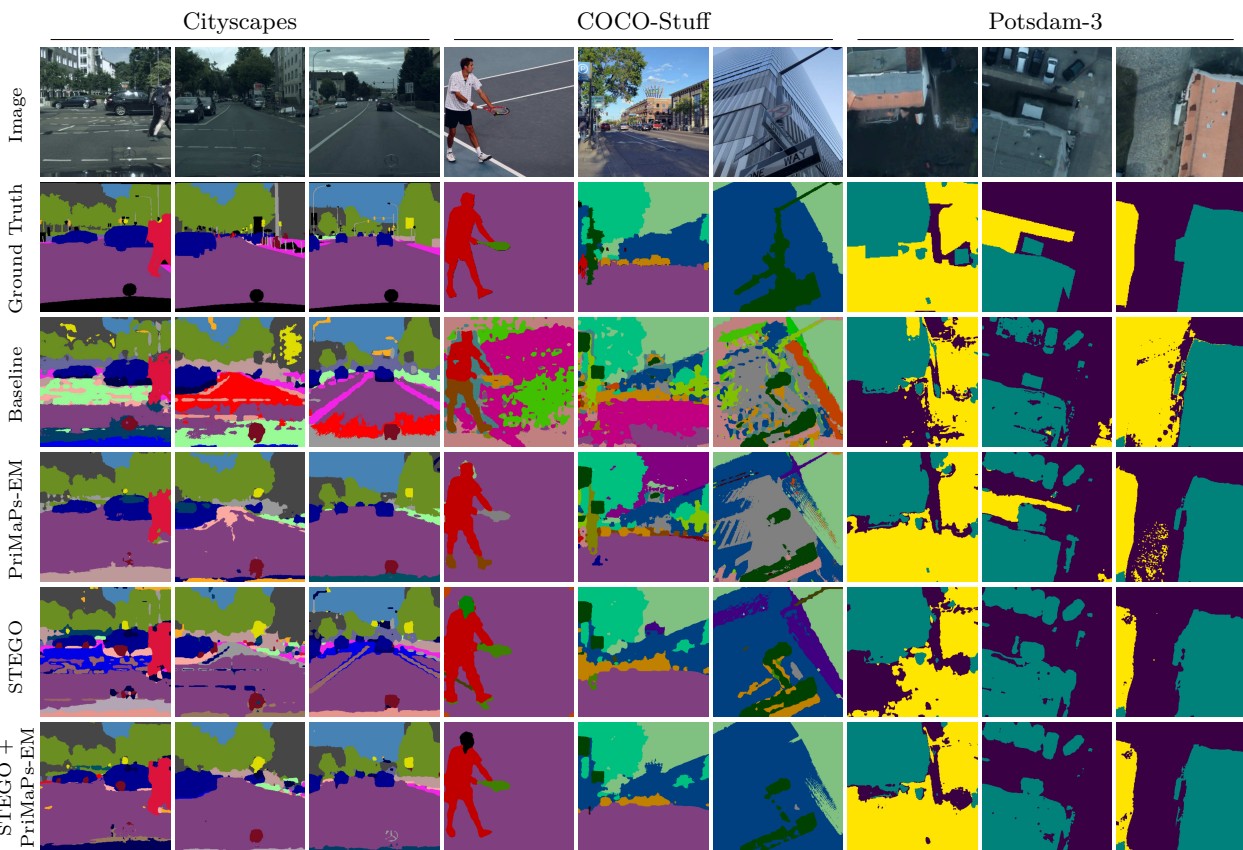

Figure 4: **Qualitative results** for the DINO ViT-B/8 baseline, PriMaPs-EM *(Ours)*, STEGO (Hamilton et al., 2022), and STEGO+PriMaPs-EM *(Ours)* for Cityscapes, COCO-Stuff, and Potsdam-3. Our method produces locally more consistent segmentation results reducing overall misclassification compared to the corresponding baseline.

local consistency of the segmentation and reduced mis-classification. The comparison with STEGO as a baseline exhibits a similar trend. For further examples and comparisons with HP, please refer to Appendix B.2.

**Limitations.** One of the main challenges is to distinguish between classes that happen to share highly similar SSL feature representations. This is hardly avoidable if the feature representation is fixed, as was the case here and in previous work (Hamilton et al., 2022; Seong et al., 2023). Another limitation across existing unsupervised semantic segmentation approaches is the limited spatial resolution. This limitation comes from the SSL training objectives (Caron et al., 2021; Oquab et al., 2024), which are image-level rather than pixel-level. As a result, we can observe difficulties in segmenting very small, finely resolved structures.

## 5 Conclusion

We present PriMaPs, a novel dense pseudo-label generation approach for unsupervised semantic segmentation. We derive lightweight mask proposals directly from off-the-shelf self-supervised learned features, leveraging the intrinsic properties of their embedding space. Our mask proposals can be used as pseudo labels to effectively fit global class prototypes using moving average stochastic EM with PriMaPs-EM. Despite the simplicity, PriMaPs-EM leads to a consistent boost in unsupervised segmentation accuracy when applied to a variety of SSL features or orthogonally to current state-of-the-art unsupervised semantic segmentation pipelines, as shown by our results across multiple datasets.

## Acknowledgments

This project is partially funded by the European Research Council (ERC) under the European Union's Horizon 2020 research and innovation programme (grant agreement No. 866008) as well as the State of Hesse (Germany) through the cluster projects "The Third Wave of Artificial Intelligence (3AI)" and "The Adaptive Mind (TAM)".

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

# A Further Analysis

In this appendix, we provide more detailed insights into PriMaPs-EM beyond the scope of the main paper.

## A.1 Nearest-Neighbor Anchoring

As described in Sec. 3 of the main paper, PriMaPs are anchoring the first principal component in each iteration of the mask proposal generation. Corresponding to the quantitative findings (*cf.* Tab. 4a), here we additionally analyze this qualitatively. Fig. 5 shows similarity maps of all features to the iteratively computed first principal component (Similarity 1st PC) as well as to the respective nearest-neighbor feature (Similarity 1st PC NN) for example images from Cityscapes (Cordts et al., 2016), COCO-Stuff (Caesar et al., 2018), and Potsdam-3 (ISPRS). We observe that the originally computed principal direction can have high similarities with multiple semantic concepts in an image. Hence, finding a suitable threshold that isolates a single main concept is difficult.

However, using the nearest image feature to the principal component as an anchoring element helps to circumvent high similarity values to multiple visual concepts. For instance, in the first example for Cityscapes, the similarity map for the first principal component has high similarities to both "building" and "vegetation". In contrast, the similarity map for the nearest neighbor of the first principal component results in high similarity to the class "vegetation" only. Consequently, a suitable mask proposal can be obtained through thresholding. We observe this particularly for the Cityscapes dataset and some examples of COCO-Stuff. Further, in cases where the similarity map is already localized to one visual concept, anchoring merely leads to a change in the similarity values without changing the shape of the thresholded proposal.

## A.2 Ablation of the Threshold

In the spirit of unsupervised learning, our method effectively has only a single additional hyperparameter – the threshold $\psi$. Furthermore, this parameter can be set simply by examining the mask proposal as detailed next. In addition, it should be noted that we keep this parameter unchanged for all backbone models and datasets, which further emphasizes the generalizability of our method. As described in Sec. 3 of the main paper, the threshold $\psi$ is used to remove noise in the similarity masks and to localize the optimization signal. In Fig. 6, qualitative examples of PriMaPs mask proposals are visualized for DINO ViT-B/8 on Cityscapes, COCO-Stuff, and Potsdam-3. We show mask proposals for $\psi = 0.3$, $\psi = 0.4$, and $\psi = 0.5$. Visually, it can be observed that meaningful masks are produced for all thresholds. Especially those for threshold 0.3 and 0.4 align very well with the semantic content in the images. While $\psi = 0.3$ seems to provide better mask proposals for COCO-Stuff and Potsdam-3, a problem arises with Cityscapes. Here, the mask proposals contain several semantic concepts and spatially small objects in a large mask (*cf.* the second Cityscapes example, where the mask of the bush in the left half of the image covers both the traffic sign in the foreground, parts of the house in the background, as well as the sky). Since this would lead to a poor optimization signal and the masks of the other two datasets using $\psi = 0.4$ seem visually appealing, the threshold is set to 0.4 in all other experiments.

To further shed light on these qualitative observations, we apply PriMaPs-EM for the scenario described above and vary the threshold from $\psi = 0.2$ to $\psi = 0.6$ with a step size of 0.1. We perform this with the DINO ViT-B/8 backbone for Cityscapes, COCO-Stuff, and Potsdam-3 and show the mIoU in Fig. 7. The quantitative results reflect our qualitative conclusion of the threshold well, and show the trade-off between better segmentation accuracy on COCO-Stuff and Potsdam-3 for a lower threshold and vice versa for Cityscapes. In numbers, a threshold of $\psi = 0.5$ instead of $\psi = 0.4$ for Cityscapes would lead to an additional gain of 1.4 % mIoU, but also to slight losses on COCO-Stuff of 0.8 %. Additionally, this results in more mask proposal iterations. We conclude that the determined threshold $\psi = 0.4$ appears to be reasonable. Even if better results could be achieved for some backbone-dataset combinations with individually set thresholds, we consider a fixed threshold that generalizes well across all scenarios to be sensible. We would like to emphasize that this experiment was conducted solely for this single backbone and serves only to validate the qualitative judgement from above. Importantly, we did not determine the hyperparameters based on the evaluation sets.

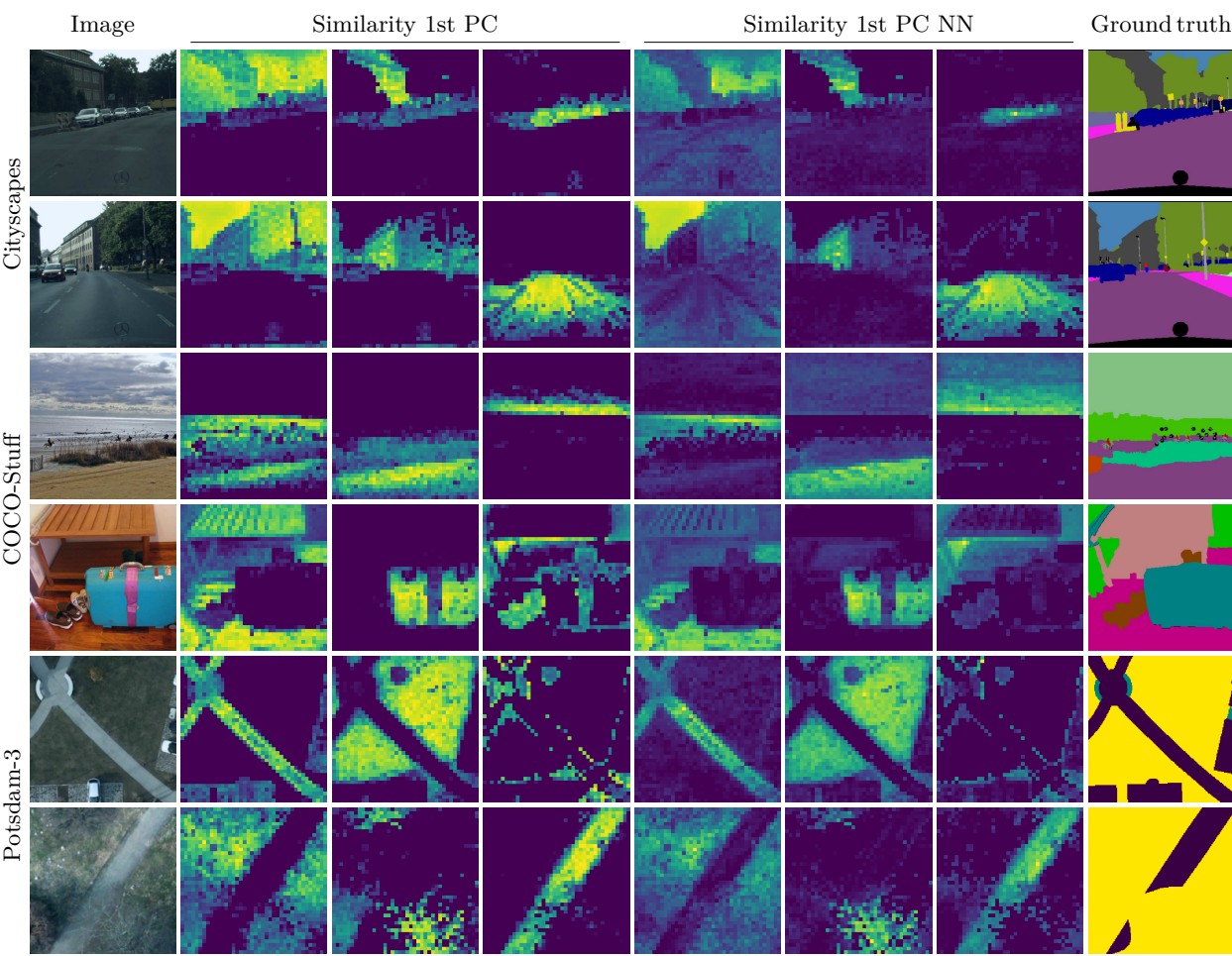

Figure 5: **Nearest neighbor anchoring of the principal direction in PriMaPs**. Image, ground-truth label, and the first three similarity maps with respect to the principal direction *(left)* and their nearest neighbor *(right)* for all three datasets using DINO ViT-B/8. Anchoring localizes the signal for principal directions with high similarities to multiple visual concepts.

## A.3 Ablation of the Stopping Criterion

Our method iteratively decomposes images into mask proposals. Hence, a stopping criterion is essential to decide whether a sufficient number of masks has been generated for a particular image. This allows us to divide different images into a different number of masks. We use the percentage of features assigned to masks. Specifically, the iterative PriMaPs process stops when 95% of image features are assigned to masks. While setting this value as high as possible to cover the entire image with masks might seem intuitive, not all features can be assigned reasonably due to the inherent noise in the self-supervised feature representation. Setting the value too high would result in masks with very few features in the final iterations. We chose a stopping criterion of 95%, which we confirmed quantitatively to yield good results across all datasets as shown in Fig. 8. The relatively low mIoU for the Potsdam-3 dataset with smaller stopping criteria results from a few initial mask proposals covering large areas, while the proposals in the final iterations represent finer details.

## A.4 Ablation of the Image Augmentations

In addition to the Tab. 4b, we analyze the photometric augmentations used in PriMaPs-EM in greater detail. As can be seen, the additional augmentation leads to slightly improved performance in terms of metrics. We used a combination of standard augmentations as well as the standard torchvision implementation and did

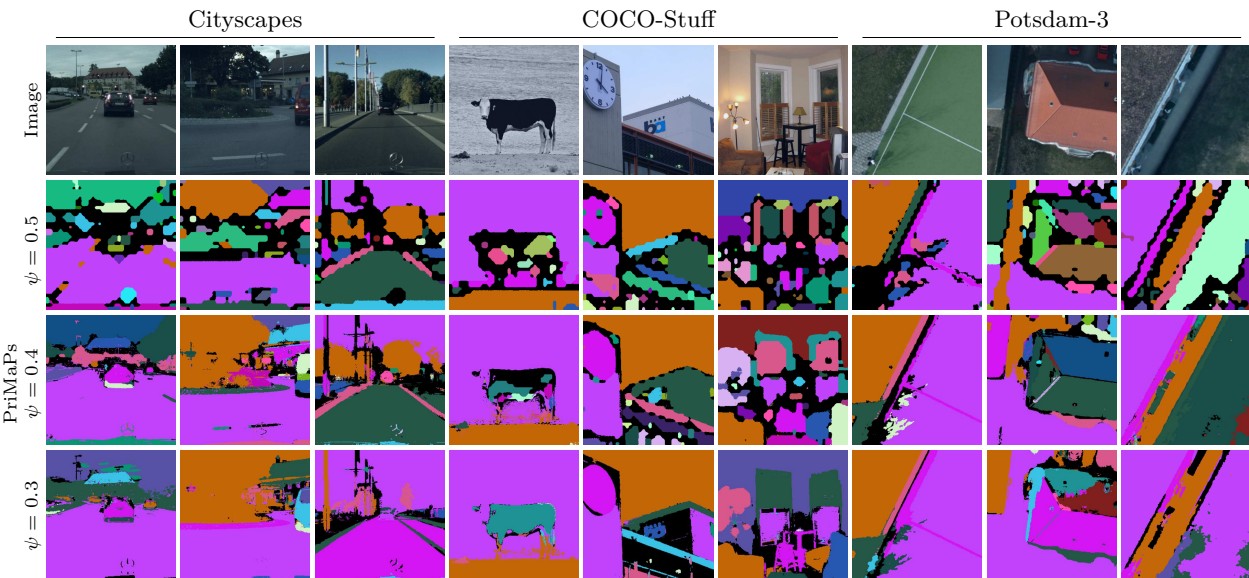

Figure 6: **Qualitative threshold $\psi$ ablation** with DINO ViT-B/8 using different $\psi$ values for PriMaPs mask-proposal generation on all three datasets. The hyperparameter exhibits favorable stability properties.

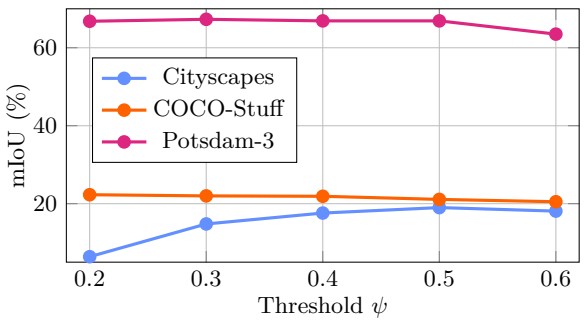

Figure 7: **Quantitative threshold $\psi$ ablation** with DINO ViT-B/8 using different $\psi$ values for PriMaPs mask-proposal generation on all three datasets. The single hyperparameter of our method exhibits favorable stability properties.

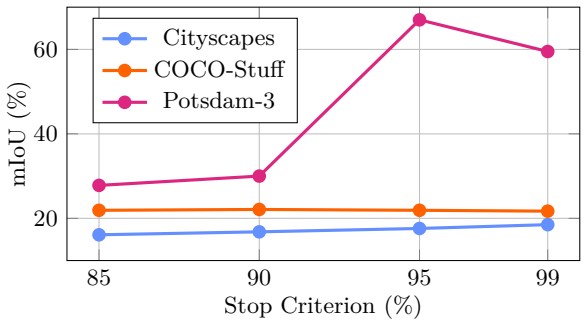

Figure 8: **Quantitative stopping-criterion ablation** with DINO ViT-B/8 using different stopping criteria for PriMaPs mask proposal generation on all three datasets.

not tune any parameters. Tab. 6 provides a contribution breakdown of the used augmentation techniques (grayscaling, Gaussian blur, color jitter) on COCO-Stuff using DINO ViT-B/8. Each of the three photometric augmentations contributes to the method's performance in terms of downstream metrics.

## A.5 Ablation on the Number of Pseudo Classes $K$

Following previous works (Ji et al., 2019; Cho et al., 2021; Hamilton et al., 2022; Seong et al., 2023) in unsupervised semantic segmentation, generally the number of pseudo classes $K$ is set to the number of annotated semantic classes in the dataset. This is done to evaluate the performance in terms of downstream metrics on every semantic class after matching the predicted pseudo-class IDs with the ground-truth classes using Hungarian matching. Using a $K$ smaller than the number of ground-truth classes is hard to evaluate and compare as it results in ground-truth classes that need to be ignored by the metric. However, it is possible to have more pseudo classes than ground-truth classes by assigning multiple pseudo classes to a single ground-truth class. We conduct an experimented in this over-segmentation setup, *i.e.* predicting $K$ categories, where $K$ is larger than the number of ground-truth classes. We realize the multi-to-one matching

Table 6: **Ablation study** analyzing design choices and components of PriMaPs-EM for COCO-Stuff using DINO ViT-B/8 including a breakdown of the image augmentations.

| Method | Acc | mIoU |
|---|---|---|
| Baseline (Caron et al., 2021) | 38.8 | 15.7 |
| + PriMaPs pseudo label | 38.8 | 18.0 |
| + EMA | 45.0 | 20.2 |
| + Augment (grayscaling) | 45.3 | 20.4 |
| + Augment (grayscaling, Gaussian blur) | 45.8 | 20.2 |
| + Augment (grayscaling, Gaussian blur, color jitter) | 46.0 | 20.4 |
| + CRF ($\equiv$ PriMaPs-EM) | 48.4 | 21.9 |

Table 7: **Over clustering** with PriMaPs-EM for COCO-Stuff using DINO ViT-B/8.

| Method | Acc | mIoU |
|---|---|---|
| PriMaPs-EM (K=27; 100%) | 48.5 | 21.9 |
| PriMaPs-EM (K=40; 150%) | 53.1 | 23.2 |

by applying Hungarian matching first and subsequently matching the remaining pseudo-class IDs based on their highest correspondence to the ground-truth classes. In Tab. 7, we report the unsupervised semantic segmentation results for COCO-Stuff with DINO ViT-B/8 once using $K = 27$, which matches the number of ground-truth classes, and for $K = 40$. Over clustering leads to better performance in terms of metrics, as multiple pseudo classes represent a single ground-truth class. Overall, estimating the number of pseudo classes or semantic concepts in a dataset in an unsupervised manner represents an intriguing direction for future research.

## B Further Experiments

This appendix provides insights beyond the experiments and ablations shown in Sec. 4.

### B.1 Class-Level Quantitative Analysis

To gain a deeper understanding of PriMaPs-EM, we assess the segmentation accuracy in terms of IoU for individual classes. Additionally, we present the confusion matrices among the semantic classes for the DINO ViT-B/8 Baseline, PriMaPs-EM, STEGO, and STEGO+PriMaPs-EM for the COCO-Stuff dataset in Fig. 9. Generally, we can observe that for both the DINO and STEGO baseline, for certain classes (*e. g.*, "Appliance", "Indoor", "Kitchen") the discovered unsupervised class concept does not correlate with human-defined semantic classes. This suggests that the respective backbone feature representation may already be hard to separate. Furthermore, some of the 27 intermediate COCO-Stuff classes merge visually distinct concepts. For instance, the class "indoor" combines "hairbrush", "toothbrush", "hair dryer", "teddy bear", "scissors", "vase", "clock", and "book". We see this assumption partially confirmed by analyzing the class IoUs of linear probing of the DINO features. For the problematic classes, the linear probing IoUs are in the range of approx. 16 %–30 %, whereas for the other classes the IoU is 50 % and higher. We conclude that if it is already difficult to linearly distinguish the classes based on the backbone features, our method can hardly improve upon this.

However, for classes meeting this requirement, our method clearly boosts class IoUs. In rare cases (*e. g.*, STEGO comparison to STEGO+PriMaPs-EM for classes "Outdoor" and "Sports"), there is a decrease in one class IoU with PriMaPs-EM while another class IoU increases. This can occur if the change in the prototype representation results in a change of the Hungarian matching for evaluation, though this is rarely observed. In terms of class confusion, the model's predictions align with the ground-truth labels. Furthermore, the existing confusions are reasonable. For instance, when using STEGO, confusions of the two "food" mid-level classes emerge. Overall, Fig. 9a indicates that our method either enhances or at least maintains the

Figure 9: **COCO-Stuff – Comparison of the segmentation performance for individual classes** in terms of IoU (in %) *(a)* and class confusion for the DINO ViT-B/8 Baseline *(b)*, PriMaPs-EM *(c)*, STEGO *(d)*, and STEGO + PriMaPs-EM *(e)*. Overall, PriMaPs-EM preserves or boosts the individual class IoU across most classes and moderately reduces class confusion.

(a) Class IoUs (in %) for DINO ViT-B/8 Baseline, PriMaPs-EM *(Ours)*, STEGO, and STEGO+PriMaPs-EM *(Ours)*.

| | Electronic | Appliance | Food | Furniture | Indoor | Kitchen | Accessory | Animal | Outdoor | Person | Sports | Vehicle | Ceiling | Floor | Food | Furniture | Rawmaterial | Textile | Wall | Window | Building | Ground | Plant | Sky | Solid | Structural | Water |
|---|---|---|---|---|---|---|---|---|---|---|---|---|---|---|---|---|---|---|---|---|---|---|---|---|---|---|---|
| Baseline | 0.0 | 0.0 | 0.0 | 0.2 | 0.0 | 0.0 | 0.0 | 15.3 | 0.1 | 58.3 | 0.5 | 9.1 | 23.7 | 3.1 | 2.2 | 8.3 | 12.2 | 0.2 | 30.7 | 11.7 | 31.9 | 21.5 | 33.8 | 77.4 | 15.0 | 27.1 | 45.1 |
| *Ours* | 0.0 | 0.2 | 0.0 | 0.8 | 0.0 | 0.0 | 0.0 | 21.4 | 4.6 | 68.7 | 0.5 | 12.6 | 36.1 | 44.4 | 0.0 | 12.1 | 14.0 | 1.1 | 45.7 | 15.9 | 38.1 | 33.3 | 42.9 | 69.1 | 29.1 | 38.9 | 61.4 |
| STEGO | 0.0 | 0.3 | 0.3 | 13.7 | 0.1 | 0.0 | 0.7 | 74.1 | 0.1 | 61.7 | 10.9 | 40.7 | 36.3 | 30.2 | 38.8 | 22.5 | 15.6 | 11.0 | 36.0 | 2.7 | 51.8 | 44.1 | 50.2 | 82.9 | 19.8 | 37.8 | 58.8 |
| STEGO+*Ours* | 0.2 | 0.0 | 0.0 | 14.1 | 0.2 | 0.0 | 0.3 | 76.8 | 10.9 | 64.3 | 0.9 | 53.8 | 48.2 | 31.9 | 39.4 | 25.4 | 16.0 | 19.7 | 36.1 | 0.8 | 55.6 | 44.9 | 51.2 | 83.8 | 27.4 | 36.2 | 62.6 |

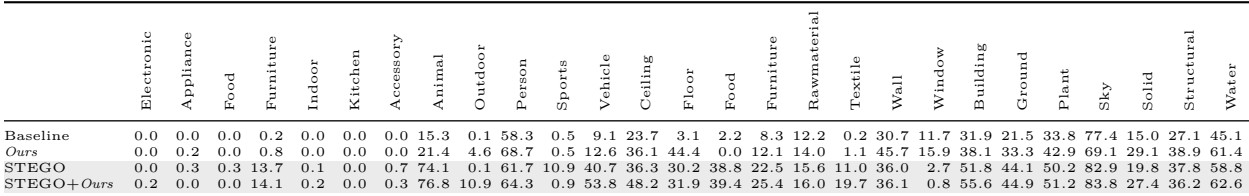

(b) Confusion matrix for the Baseline

(c) Confusion matrix for PriMaPs-EM

(d) Confusion matrix for STEGO

(e) Confusion matrix for STEGO + PriMaPs-EM

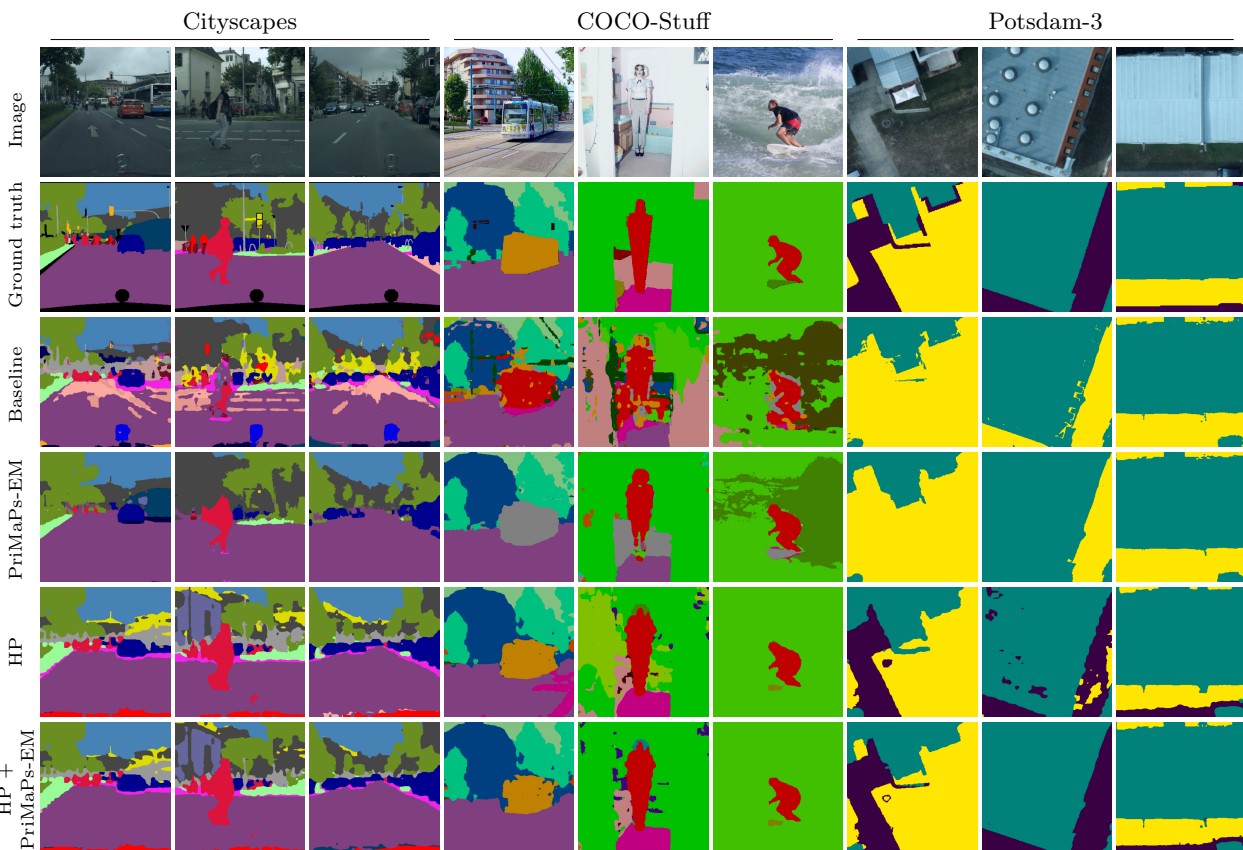

Figure 10: **Qualitative results** for the DINO ViT-S/8 Baseline, PriMaPs-EM *(Ours)*, HP (Seong et al., 2023), and HP (Seong et al., 2023)+PriMaPs-EM *(Ours)* for all three datasets. Our method produces locally more consistent segmentation results, reducing misclassification.

segmentation performance per class in terms of IoU regardless of the backbone model. Additionally, our method aids in reducing the confusion among classes.

## B.2 Qualitative Comparison to HP

Similar to the qualitative comparison in Fig. 4 in the main paper, we aim to compare PriMaPs-EM with HP (Seong et al., 2023). We present qualitative examples for the baseline, PriMaPs-EM, HP, and the combination of HP and PriMaPs-EM in Fig. 10. These qualitative examples align with the findings from the quantitative results in the main paper (*cf.* Tabs. 1 to 3). It is evident that our method produces locally more consistent and less noisy results compared to both baselines. Despite the already impressive qualitative results of HP on COCO-Stuff, our method excels in correcting misclassifications and achieving better segmentation of object boundaries. We observe one limitation, particularly with the DINO ViT-S/8 baseline, both independently and in conjunction with PriMaPs-EM for the Potsdam-3 dataset. In this case, the semantic concept of "street" is not recognized and is consequently rarely predicted.

## B.3 Comparison to HP using DINOv2

Current state-of-the-art methods Hamilton et al. (2022); Seong et al. (2023) for unsupervised segmentation do not provide experiments using DINOv2 features. To be able to compare with previous methods, we train HP using DINOv2 for the comparison in Tab. 8. We strictly follow the training schedule and hyperparameters used in the original implementation for all respective datasets. HP does not generalize well to the DINOv2 features and hardly keeps up with the strong baseline. PriMaPs-EM moderately but consistently improves

Table 8: **Comparing PriMaPs-EM *(Ours)* to HP using DINOv2** for the Cityscapes (ViT-B/14), COCO-Stuff (ViT-B/14), and Potsdam-3 (ViT-S/14) datasets. We report Accuracy and mean IoU (in %) for unsupervised probing.

| Method | *Cityscapes* | | *COCO-Stuff* | | *Potsdam-3* | |
|---|---|---|---|---|---|---|
| | **Acc** | **mIoU** | **Acc** | **mIoU** | **Acc** | **mIoU** |
| DINOv2 Baseline (Oquab et al., 2024) | 49.5 | 15.3 | 44.5 | 22.9 | 82.4 | 69.9 |
| + HP (Seong et al., 2023) | 67.9 | 15.9 | 48.9 | 19.8 | 79.4 | 65.7 |
| + PriMaPs-EM | 71.6 | **19.0** | 46.4 | **23.8** | **83.1** | **71.0** |
| + HP (Seong et al., 2023) + PriMaPs-EM | **74.3** | 16.6 | **49.3** | 20.2 | 79.6 | 66.0 |

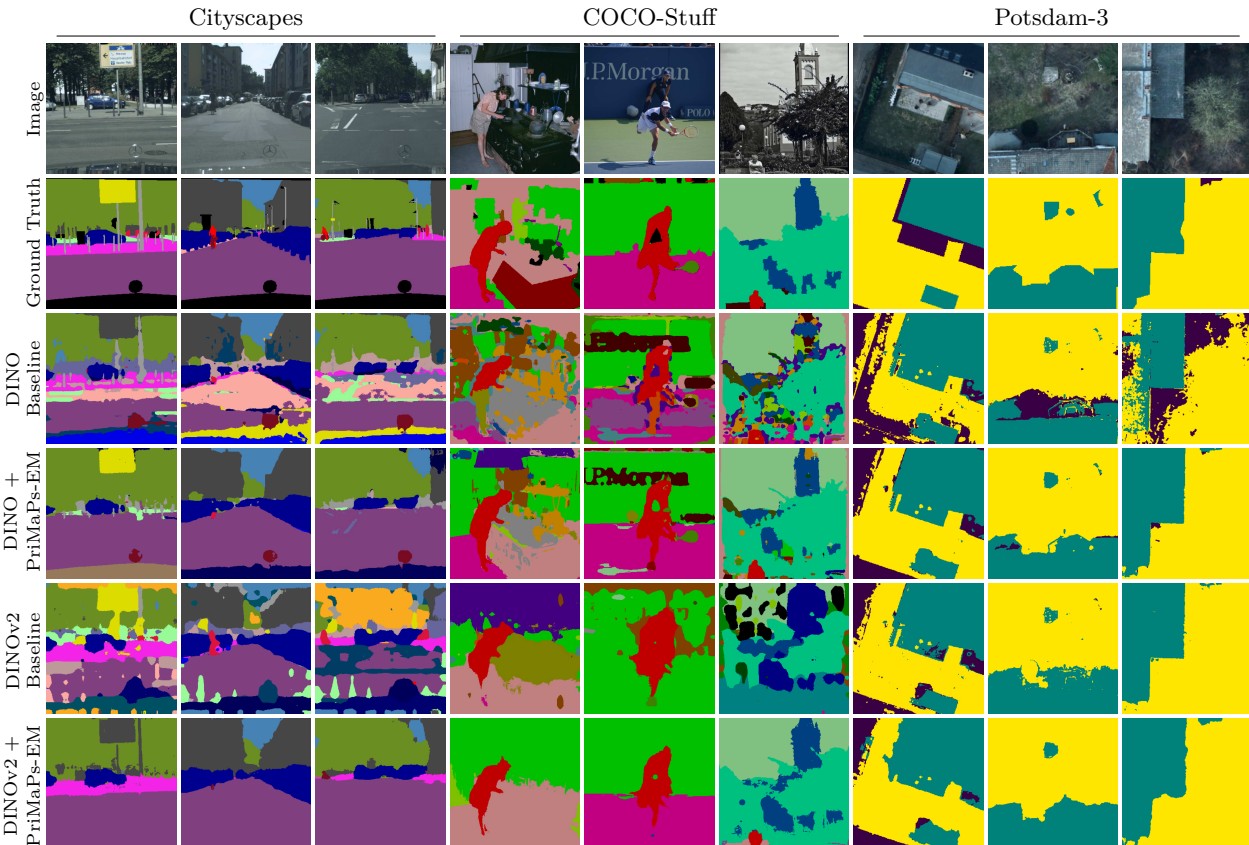

Figure 11: **Qualitative comparison** of DINO and DINOv2 using ViT-B/8 and ViT-B/14 respectively. We show the DINO Baseline, DINO+PriMaPs-EM *(Ours)*, DINOv2 Baseline and DINOv2+PriMaPs-EM *(Ours)* for Cityscapes, COCO-Stuff, and Potsdam-3.

upon both DINOv2 and HP across all datasets and metrics using the identical PriMaPs-EM hyperparameters we use across all other experiments in this work.

## B.4   Qualitative Comparison of DINO and DINOv2

Our experiments could not identify any clear differences between DINOv1 and DINOv2 and the different transformer architectures in terms of the downstream task performance of unsupervised semantic segmentation. Both methods provide excellent feature embeddings and similar quantitative results as baselines. When applying PriMaPs-EM, DINOv2 often provides slightly better quantitative results but coarser segmentation

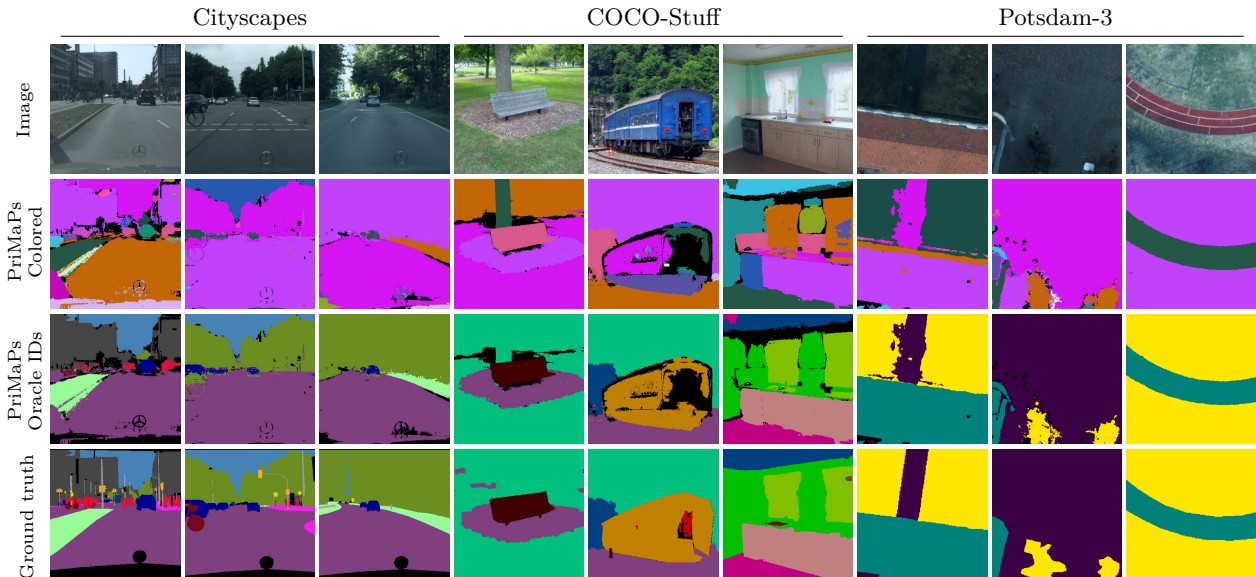

Figure 12: **Qualitative PriMaPs examples** using DINO ViT-B/8 for Cityscapes, COCO-Stuff, and Potsdam-3. *PriMaPs Colored* – each mask proposal is visualized in a different color. *PriMaPs Oracle class IDs* – each mask is colored in the corresponding ground-truth class color.

masks due to the larger patches (*cf.* Tabs. 1 to 3). We visualize both the DINO and DINOv2 baselines and with PriMaPs-EM in Fig. 11.

### B.5 Qualitative PriMaPs Pseudo-Label Examples

Following the quantitative assessment of the pseudo labels in Sec. 4.3, Fig. 12 visualizes qualitative examples of the PriMaPs mask proposals and pseudo labels. We visualize individual masks, each in a different color (*PriMaPs Colored*). We also display oracle pseudo labels, assigning each mask a color based on the ground-truth label (*PriMaPs Oracle class IDs*). We observe that the mask proposals align well with the ground-truth labels across all three datasets, generalizing across three distinct domains. PriMaPs effectively partitions images into semantically meaningful masks.

### B.6 Failure Cases

Finally, we would like to discuss observed failure cases of PriMaPs-EM. Fig. 13 shows examples of failure cases occurring in the segmentation predictions as well as failure examples for PriMaPs pseudo labels. For PriMaPs-EM, we observe misclassifications, such as in Cityscapes where buses are often partially segmented as the class "car" or that cobblestone is misclassified as "sidewalk". For COCO-Stuff, shadows and structures influence the segmentation predictions and confusions also occur (see "ground" and "floor" in example two). For Potsdam-3, we observe that vehicles and road markings are sometimes erroneously attributed to the class "building" instead of "road". For PriMaPs pseudo labels, we identify two main sources of error. First, small objects in cluttered images are sometimes assigned to larger neighboring masks, a phenomenon that can be attributed to the limited backbone feature resolution. This is particularly noticeable for Cityscapes, where the center horizontal area is often detailed (*cf.* Fig. 12 "Pole", "Traffic Light", "Traffic Sign"). Second, PriMaPs sometimes oversegment images containing a large foreground object as seen in the COCO-Stuff example. Similarly, different visual appearances of the same semantic class can lead to multiple masks (Potsdam-3). Despite these observations, the simple PriMaPs provide promising mask proposals that correspond well with the ground-truth label.

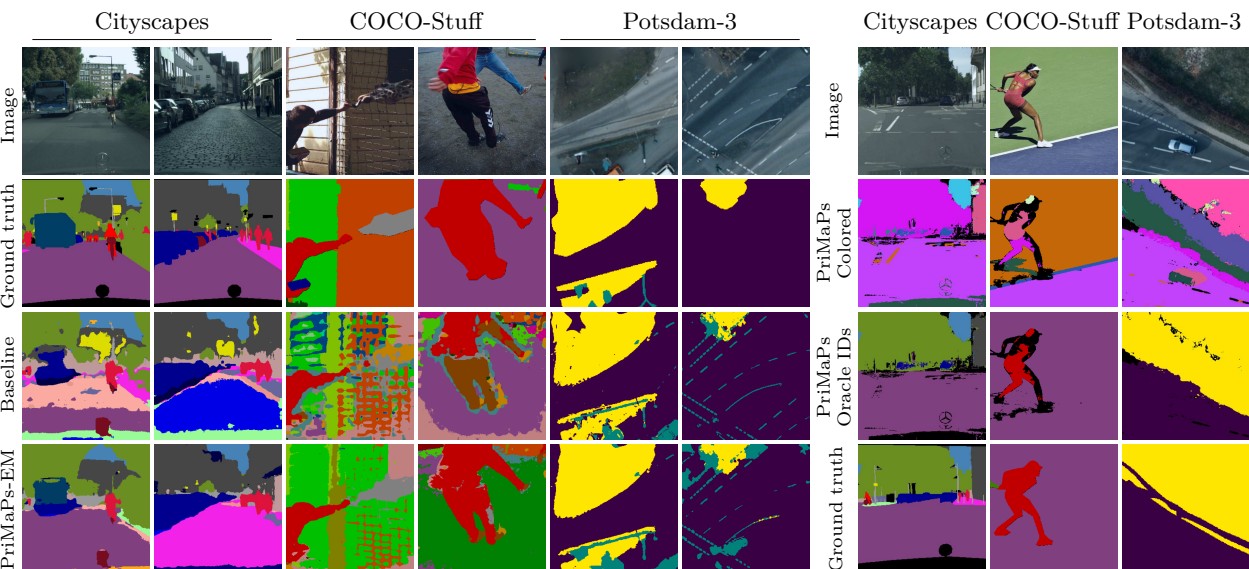

Figure 13: **Failure cases** for the PriMaPs-EM segmentation *(left)* as well as PriMaPs pseudo labels *(right)* using DINO ViT-B/8 for Cityscapes, COCO-Stuff, and Potsdam-3.

## C  Implementation

Since all significant high-level implementation details and hyperparameters have been addressed in the main paper, this section addresses only few remaining details. Both the code and models are publicly available at `https://github.com/visinf/primaps`. Our work is implemented in PyTorch (Paszke et al., 2019). We build up on the code of Ji et al. (2019), Van Gansbeke et al. (2021) and Hamilton et al. (2022).

### C.1  Backbone Models

For each backbone model, we use the corresponding original implementation. Specifically, for DINO (Caron et al., 2021) and DINOv2 (Oquab et al., 2024), we utilize the PyTorch Hub implementation. In the case of STEGO (Hamilton et al., 2022) and HP (Seong et al., 2023), we integrate the respective original implementations into our framework.

### C.2  Computational Requirements

We perform all experiments on a single NVIDIA A6000 GPU. The PriMaPs-EM optimization runtime varies based on dataset size and the ViT architecture used as the backbone, with ViT-B/8 exhibiting the longest runtime and highest memory consumption. Specifically, using DINO ViT-B/8, PriMaPs-EM requires about 2.5 hours for Cityscapes and Potsdam, and approximately 4 hours for COCO-Stuff, utilizing about 30GB of memory. For the lightest backbone, DINOv2 ViT-S/14, optimization times decrease to about 1.5 hours for Cityscapes and Potsdam-3, and around 2.5 hours for COCO-Stuff, with a memory usage of about 20GB. DINO ViT-S/8 and DINOv2 ViT-B/14 lie in between.

### C.3  Datasets

We close with some further details regarding the datasets used.

**Cityscapes**  (Cordts et al., 2016) is an ego-centric street-scene dataset containing 5000 high-resolution images with $2048 \times 1024$ pixels. It is split into 2975 train, 500 val, and 1525 test images. Following previous work (Ji et al., 2019; Cho et al., 2021; Yin et al., 2022; Hamilton et al., 2022; Seong et al., 2023), evaluation is conducted on the 27 classes setup using the val split.

**COCO-Stuff** (Caesar et al., 2018) is a dataset of everyday life scenes containing 80 things and 91 stuff classes. Following previous work (Ji et al., 2019; Cho et al., 2021; Hamilton et al., 2022; Yin et al., 2022; Li et al., 2023; Seong et al., 2023), we use a reduced variant by Ji et al. (2019) containing 49629 train and 2175 test images. Hereby, all images consist of at least 75 % stuff pixels. The dataset is evaluated on the 27 classes setup.

**Potsdam-3** (ISPRS) is a remote sensing dataset consisting of 8550 RGBIR satellite images with $200 \times 200$ pixels, which is split into 4545 train and 855 test images, as well as 3150 additional unlabeled images. In our experiments, the 3-label variant of Potsdam is evaluated and the additional unlabeled images are not used.

