# OpenReview forum: "Boosting Unsupervised Semantic Segmentation with Principal Mask Proposals"
_TMLR — Accepted by TMLR_

### Review · Reviewer_BQDo · 2024-05-26

**Summary Of Contributions:**

The paper presents a method called Principal Mask Proposals (PriMaPs) for unsupervised semantic segmentation. The method involves using an iterative approach to identify semantic masks using features from a pretrained self-supervised model. Based on the masks, the authors construct pseudo labels and employ a moving average stochastic expectation-maximization algorithm called PriMaPs-EM. This technique assigns a consistent semantic class to each mask proposal, ensuring global consistency across the image regions. The authors demonstrate that proposed approach is a general approach that can be applied orthogonally to existing unsupervised semantic segmentation pipelines, thereby boosting their performance. Experimental results show consistent improvements in segmentation accuracy on multiple datasets.

**Audience:**

Yes

**Claims And Evidence:**

Yes

**Requested Changes:**

See weakness part.

**Strengths And Weaknesses:**

Strengths:
1. Important field of study: Unsupervised semantic segmentation is important in the study of scene understanding.
2. Simple method: The proposed method is simple, enhancing its accessibility and implementation.
3. Impact of proposed solution: The proposed solution can be potentially impactful as it is an efficient strategy to improve existing models.

Weaknesses:
1. Writing issues: Certain details are missing from the paper, make it not less straightforward to follow. For instance, does $y$ refer to the prediction in pixel space or feature space. Why dot product is applied to the encoded feature and prototype, isn't the prototype the eigenvectors of the encode features. The paper doesn't clearly state how their method can be used orthogonally with existing method, for current version, it seems like the method itself can be a standalone method for unsupervised segmentation.
2. Class ambiguity: The approach may struggle to distinguish between classes that share similar SSL feature representations. This can lead to misclassifications, particularly in complex scenes with multiple similar objects.
3. Generalisation ability: The method relies on fixed SSL feature representations, which may limit its ability to fully adapt to specific segmentation tasks. If the SSL features are not well-suited for a particular dataset (For example, DINO for CityScape dataset), the segmentation performance could be suboptimal.

In general, the paper writing requires further reviews for more details and clarification. While I think the idea might have certain limitations, the proposed solution is interesting and novel, therefore I would like to see authors' reponses towards the aforementioned concerns.

---

> ### Author Response · Authors · 2024-06-30
> **Response to Reviewer BQDo**
>
> Thank you for the valuable feedback.
>
>
> We apologize for any confusion regarding the method section. We will clarify the open points here and revise the paper accordingly.
>
>
> ## Q1.1: Does $y$ refer to the prediction in pixel space or feature space?
>
> $y$ refers to the prediction in the pixel space.
> Similar to STEGO and HP, we bilinearly upsample the SSL features to the pixel resolution in PriMaPs-EM (we use feature resolution for PCA in Sec. 3.1).
> We also upsample the binary PriMaPs bilinearly for CRF refinement.
> We will make this clearer in the revision.
>
>
> ## Q1.1: Why dot product is applied to the encoded feature and prototype?
>
> The dot product is efficient and represents a meaningful metric, since we are operating in the Euclidean vector space. It is equivalent to a linear layer without the bias term.
>
>
> ## Q1.2 Isn't the prototype the eigenvector of the encoded features?
>
> No, it is not. As mentioned in 3.2 PriMaPs-EM, "The class prototypes $\theta_S$ and $\theta_T$ are the K pseudo-class representations in the feature space, projecting the C-dimensional features linearly to K semantic pseudo-classes. It is a set of vectors optimized with PriMaPs-EM.
>
>
> ## Q1.3: How is the method orthogonal to existing approaches?
>
> PriMaPs and PriMaPs-EM are agnostic to the initial feature representation, be it an SSL approach (e.g. DINO) or another posthoc segmentation method (e.g. STEGO). Since we observe improvements in segmentation accuracy regardless, PriMaPs/PriMaPs-EM are orthogonal to this existing work. We will clarify this in the revision.
>
>
> ## Q2: The approach may struggle to distinguish between classes that share similar SSL feature representations.
>
> This is indeed an issue. However, it is shared with all previous work operating with frozen SSL features (e.g. STEGO, TransFGU, ACSeg, HP).
> Nevertheless, our work demonstrates that these features can perform semantic segmentation without human annotation and discover many of the semantic classes defined across three entirely distinct datasets. We analyze this in more detail in Sec. B1 of the supplemental.
>
>
> ## Q3: The method relies on fixed SSL feature representations, which may limit its ability to fully adapt to specific segmentation tasks.
>
> We agree. Indeed, the use of fixed SSL features is intentional by design. As we discuss in Sec. 1, our goal is to study the bounds of the fixed SSL features for semantic segmentation with a lightweight approach comparable to linear probing.

---

### Review · Reviewer_DBXQ · 2024-06-02

**Summary Of Contributions:**

The authors have proposed a method for unsupervised semantic segmentation that can work with the feature outputs from standard SSL-trained networks, such as DINO, or even previously supervised semantic segmentation networks.

Initially, the authors proposed PriMaPs, which iteratively identifies the first principal component, finds its 1-NN patch, and sets that as a class. It assigns all cosine similarities above a certain threshold to the same class. This process is repeated, removing these patches until 95% of the image is masked.

The second contribution builds on the first by following a process similar to EM. They generate K initial class prototypes. Using these prototypes and the backbone features, they create predictions, which are then used as pseudo labels in a batch-wise focal loss to optimize the prototypes. The new prototypes are used in a moving average to update the prototypes.

**Audience:**

Yes

**Claims And Evidence:**

Yes

**Requested Changes:**

Address the clarity issues and provide the missing ablations.

**Strengths And Weaknesses:**

The main strength of the paper lies in its generalizability, as it can be applied to the output of other patch-based methods, whether supervised or not, and enhance semantic segmentation.

As far as I know, the method is novel. The PCA-based labeling method they propose is very interesting, and I believe it can be generalized to other tasks as well.

The results are impressive. While the improvement on top of supervised methods is marginal, it is consistent and far more impressive than the larger improvement over the SSL method.

The main weaknesses are around the clarity of the paper. While most of the paper is clear, the method section is quite confusing.

In section 3.1:

1.	I would try to focus on the difference between standard PCA and the method you are using. As it is written now, it is unclear where PCA ends and your addition begins.
2.	The sub-subsection titled “Proposal post-processing” seems more fitting for section 3.2, as the CRF method is orthogonal and commonly used in different methods to improve segmentations.

In section 3.2:
General question: You moved to batches. Are these batches of images or batches of patches inside a single image? Is the method used on an entire training set (with the same semantic classes) at once, or only valid for a single image at a time?

1.	Initialization: Are you using standard PCA or your PriMaPs method? If it’s the former, please elaborate on why. If it’s the latter, please make it clearer.
2.	MA: This entire part is unclear: “We use the backbone features and momentum class prototypes θT to yield a segmentation prediction y from which pseudo labels are generated as described in Sec. 3.1.” This is not described in 3.1. In 3.1, you describe how to use feature maps, but I am not sure how the prototypes are used. Focal Loss: How exactly do you use it? I understand it is optimized according to it, but where exactly? The entire optimization process of the prototypes (apart from updating using the MA) and the way you use them is unclear.

Another question: When assigning classes, do you still use the 95% rule from the first part, or do you continue until you have k classes? Why do you have psi as a subscript in the decay factor? Is it related to the previous psi you use in 3.1?

Aside from the clarity issue, some ablations are missing for the first part. Specifically, how would using the standard K leading eigenvectors instead of the current process to get the single eigenvector at a time work? Additionally, ablating the threshold psi would be beneficial.

---

> ### Author Response · Authors · 2024-06-30
> **Response to Reviewer DBXQ**
>
> We thank the reviewer for the valuable feedback.
>
> We apologize for any confusion regarding the method section. We will clarify the open points here and revise the paper accordingly.
>
> ## Q1: Difference between standard PCA and the method in section 3.1?
>
> PriMaPs iteratively obtain the first principal component with PCA, followed by mask proposal generation (c.f. Sec. 3.1).
> The standard PCA computes all principal components at once.
> However, we found the iterative strategy to perform slightly better (see Tab. 4a).
> We will make this distinction clearer in the revision.
>
> ## Q2: The sub-subsection titled “Proposal post-processing” seems more fitting for section 3.2.
>
> To clarify, we define PriMaPs as the outcome of the entire proposal generation pipeline, including proposal post-processing. Therefore, it seems appropriate to detail the proposal post-processing in Sec. 3.1. Nevertheless, we are happy to revise in favor of a clearer presentation structure if advised by the reviewer.
>
> ## Q3: You moved to batches. Are these batches of images or batches of patches inside a single image?
>
> PriMaPs-EM uses batches of images (set to 32 in our experiments).
>
> ## Q4: Is the method used on an entire training set (with the same semantic classes) at once, or only valid for a single image at a time?
>
> PriMaPs-EM fits a fixed set of class prototypes to the entire set of training images.
>
>
> ## Q5: Initialization: Are you using standard PCA or your PriMaPs method?
>
> We use the standard PCA extracting the first K principal components (c.f. Sec. 4.1).
> This is motivated by prior work (Drineas et al., 2004; Ding \& He, 2004), which showed that the PCA subspace, spanned by principal components, is a relaxed solution to K-means clustering.
> We found this approach to provide a favorable initialization for segmentation in practice.
>
>
> ## Q6: Unclear part “We use the backbone features and momentum class prototypes ..."
>
> We agree that this could be made clearer and we will revise it as follows.
> We first compute the dot product of the dense C-dimensional image features $f$ with $K$ class prototypes contained in $\theta_T$.
> The following argmax operation produces the dominant prototype for each feature location, resulting in $K$ prototype-based binary masks.
> Taking PriMaPs from Sec. 3.1 (which do not require prototypes), we assign each mask proposal a pseudo-class ID associated to the prototype with the largest intersection.
> This process yields the final pseudo-label map P*.
>
> ## Q7: Focal Loss: How exactly do you use it?
>
> First, we construct pseudo labels using PriMaPs and the semantic segmentation prediction using the momentum class prototypes $\theta_T$. Next, we optimize the prototypes $\theta_S$ with the loss function in Eq. (10) based on the pseudo labels using gradient descent; $\theta_T$ is the moving average of $\theta_S$.
> We will make this clearer in the revision.
>
> ## Q8: When assigning classes, do you still use the $95$% rule from the first part, or do you continue until you have k classes?
>
> To generate PriMaPs we always use the $95$% stopping criterion.
> We assign prototype IDs only to PriMaPs, while ignoring the remaining pixel area (i.e. $5$%).
>
> ## Q9: Why do you have psi as a subscript in the decay factor? Is it related to the previous psi you use in 3.1?
>
> Thank you for pointing this out.
> We will remove the subscript, since it is not related to $\psi$ in Sec. 3.1.
>
> ## Q10: Ablations are missing for the first part. Specifically, how would using the standard K leading eigenvectors instead of the current process to get the single eigenvector at a time work? Additionally, ablating the threshold psi would be beneficial.
>
>
> In fact, we already include both experiments in the paper.
> We will make it more prominent in the revision.
>
> Tab. 4a reports the first experiment, where we used K leading eigenvectors instead of our iterative approach, referred to as ''PriMaPs-EM (non-iter.)''. This leads to slightly inferior results. We also implemented a very simple version ''PriMaPs-EM (naive)'', which uses the K leading eigenvectors and simply assigns the masks based on the argmax without any iterations. This leads to significantly worse results. Note that for all results, the nearest neighbor to the principal component and thresholding are still used. Below are the numbers for unsupervised semantic segmentation on COCO-Stuff using DINO ViT-B/8:
>
> | Method                 | Acc  | mIoU |
> |------------------------|-----:|-----:|
> | PriMaPs-EM (naive)     | 43.1 | 19.9 |
> | PriMaPs-EM (non-iter.) | 47.9 | 21.7 |
> | PriMaPs-EM             | 48.4 | 21.9 |
>
> For the ablation of the threshold $\psi$, we would like to refer to Sec. A2, where we analyzed it both qualitatively and quantitatively.

---

> > ### Comment · Reviewer_DBXQ · 2024-07-17
> > **reply**
> >
> > I thanks the Authors for their response. Most of my issues were around clarification of the paper, and the authors has addressed those.
> >
> > I am leaning on accept, and while the clarification here is useful, it should be reflected in the writing itself, and it should be improved before submitting the final version.

---

### Review · Reviewer_v3pd · 2024-06-17

**Summary Of Contributions:**

The paper introduces PriMaps (Principal Mask Proposal), a method to generate semantically meaningful image segmentation masks using only image representations, learned by self-supervised learning models like DINO, DINOv2. The method operates by performing PCA over learned representations and then iteratively decomposing the image into segments by computing the affinity between image features and the respective first principal component. The method further leverages conditional random fields (CRF) to refine the predicted masks around edges, and use expectation maximization alogorithm to create pseudo labels for predicted masks.

**Audience:**

Yes

**Broader Impact Concerns:**

The paper proposes a new method from generating segmentation masks of an image using only pre-trained self-supervised embeddings. There are no broader impact concerns related to the work.

**Claims And Evidence:**

Yes

**Requested Changes:**

- First, kindly address the questions mentioned above section. Add additional details where necessary.
- nitpick: - Repeated references in the related work section. Line with “Nevertheless, most of this work… “
- Try to add visualizations for the method section.

**Strengths And Weaknesses:**

### Strengths:
- The paper proposes a novel method for generating semantic image segments using only SSL representations alone.
- Unlike other unsupervised semantic segmentation methods, PriMaps introduces lightweight post-processing of SSL image representations to create segmentation masks. Since the representations are directly used, without any adaptation/finetuning, the method is comparatively more practical.

### Weaknesses:
- The method section is hard to follow. I would encourage authors to add some visualisations in method section. To elaborate more on my concerns, I have asked questions for further clarifications, kindly consider adding additional explanations where necessary.


### Questions/Clarifications:
- How expensive (time and compute-wise) is the linear optimisation (post-processing) of representations, anchor selection and EM optimization?

- Since the method is completely unsupervised (performance upper bounded by linear probing classification performance), can some sort of fine-tuning or adaption be done to  improve the performance of masks generation, and align masks with images semantically?

- How do models pre-training distributions influence the inferred semantics of segmentations on a new target dataset? As DINOv1 is pre-trained on imagenet and DINOv2 is pretrained on a large curated dataset, do these pre-training datasets induce different models in behavior?

- Heuristics for selecting K (unknown classes) within a dataset? What happens with large K and small K?
- Did you ablate image augmentations? What are the guidelines on selecting optimal augmentation setting?
- What are the heuristics behind using the nearest neighbour feature (to first principal component) as spatial anchor?
- Why/How does dominant feature (a spatial anchor) ensure consistency for localising multiple concepts within an image?
- Regarding reliability of using dominant feature ($\tilde{f}$) for spatial localisation of concepts within image, how often do the semantically same concept get split across multiple $\tilde{f}$ i.e., the same concept is explained by possibly two or more $\tilde{})?
- For “Further mask proposal”, do you still consider the first principal component and select second, third and so forth similar vectors as spatial anchors? or second, third, forth… principal components are used? Please mention this explicitly.
- As far as I understood, selection of Z is based on how much image region is left to be segmented. In the paper it says 95%, is this threshold based on some statistical distribution of segments within natural images?
- Instead of SSL method, how would a supervised, pre-trained method behave? Did authors try that?
- Could you please shed some light the selection of K and induced Z? and how intricate is there relationship?

---

> ### Author Response · Authors · 2024-06-30
> **Response to Reviewer v3pd**
>
> Thank you very much for your valuable feedback.
>
> We apologize for any confusion regarding the method section. We will clarify the open points here and revise the paper accordingly.
>
> ## Q1: The method section is hard to follow. Add some visualizations in method section.
> We included visualizations of the method in Figure 2, where Figure 2a provides a comprehensive overview of the PriMaPs-EM architecture, and Figure 2b focuses on PriMaPs and pseudo-label generation. We will improve clarity and better integrate the visualizations into the text. We are happy to visualize additional aspects of the method if specified by the reviewer.
>
> ## Q2: How expensive (time and compute-wise) is the method?
>
> Fitting class prototypes with PriMaPs-EM takes on average about 2 hours on Cityscapes on a single NVIDIA A6000 GPU (c.f. Sec. 4.1).
> The PriMaPs-EM optimization runtime varies based on dataset size and the ViT architecture used as the backbone, with ViT-B/8 exhibiting the longest runtime and highest memory consumption. Specifically, using DINO ViT-B/8, PriMaPs-EM requires about 2.5 hours for Cityscapes and Potsdam, and approximately 4 hours for COCO-Stuff, utilizing about 30GB of memory. For the lightest backbone, DINOv2 ViT-S/14, optimization times decrease to about 1.5 hours for Cityscapes and Potsdam, and around 2.5 hours for COCO-Stuff, with a memory usage of about 20GB. DINO ViT-S/8 and DINOv2 ViT-B/14 lie in between.
>
> ## Q3: Can some sort of supervised fine-tuning or adaptation be done to improve the performance?
>
> Supervised fine-tuning / adaptation techniques for further improving the segmentation accuracy are conceivable.
> While interesting, it would be orthogonal to our contribution and require a completely different experimental protocol and baselines.
>
> ## Q4: Effect of different pre-training distributions on backbone model?
>
> Our experiments could not identify any clear differences between DINOv1 and DINOv2 and the different transformer architectures. Both models provide excellent self-supervised representations. Nevertheless, we could add qualitative examples comparing DINOv1 and DINOv2 to the paper supplemental.
>
> ## Q5: Heuristics for selecting K (unknown classes) within a dataset? What happens with large K and small K?
>
> Following previous works, K is set to the number of annotated semantic classes in the dataset. This is done to evaluate the performance in terms of downstream metrics after matching the predicted pseudo-class IDs with the ground truth classes using Hungarian matching. Using a K smaller than the number of ground truth classes is hard to evaluate and compare as it results in ground truth classes that need to be ignored by the metric. Nevertheless, we find the idea interesting and experimented with over-segmentation, i.e. predicting K categories, where K is larger than the number of ground-truth classes. We realize the multi-to-one matching by applying Hungarian matching first and subsequently matching the remaining pseudo-class IDs based on their highest correspondence to the ground truth classes. In the following, we report the unsupervised semantic segmentation results for COCO-Stuff with DINO ViT-B/8 once using K=27, which matches the number of ground truth classes and for K=40. This approach leads to better performance in terms of metrics, as multiple pseudo-classes are matched to a single ground truth class.
>
> | Method                          | Acc  | mIoU |
> |---------------------------------|-----:|-----:|
> | PriMaPs-EM (K=27; 100\%)        | 48.5 | 21.9 |
> | PriMaPs-EM (K=40; ~150\%)       | 53.1 | 23.2 |
>
> ## Q6: Did you ablate image augmentations? What are the guidelines on selecting optimal augmentation setting?
>
> We ablate our method with and without augmentations in Table 4b. As can be seen, the additional augmentation leads to slightly improved performance in terms of metrics. We used a combination of standard augmentations as well as the standard torchvision implementation and did not tune any parameters.
> Below, we provide further contribution breakdown of the used augmentation techniques (grayscaling, Gaussian blur, color jitter) on COCO-Stuff using DINO ViT-B/8.
>
> | Method                                               | Acc  | mIoU |
> |------------------------------------------------------|-----:|-----:|
> | Baseline (Caron et al., 2021)                        | 38.8 | 15.7 |
> | + PriMaPs pseudo label                               | 38.8 | 18.0 |
> | + EMA                                                | 45.0 | 20.2 |
> | + Augment (grayscaling)                              | 45.3 | 20.4 |
> | + Augment (grayscaling, Gaussian blur)               | 45.8 | 20.2 |
> | + Augment (grayscaling, Gaussian blur, color jitter) | 46.0 | 20.4 |
> | + CRF (== PriMaPs-EM)                                | 48.4 | 21.9 |

---

> ### Author Response · Authors · 2024-06-30
> **Response to Reviewer v3pd**
>
> ## Q7: Heuristics behind using the nearest neighbor feature as spatial anchor?
>
> Due to the nature of PCA, the principal component is not necessarily an element of the vector set from which it was computed, so the choice to localize it with its nearest neighbor is natural. We analyze this in more detail in the supplemental Sec. A.1 Nearest Neighbor Anchoring.
>
> ## Q8: How does dominant feature (a spatial anchor) ensure consistency for localizing multiple concepts within an image?
>
> Using the nearest image feature to the principal component as an anchoring element helps to circumvent high similarity values to multiple visual concepts when generating PriMaPs. It has barely any influence in terms of attending to multiple instances of the same semantic concept in a single image. This can be observed in, e.g., Figure 5, the upper image for trees or cars.
>
> ## Q9: How often do the semantically same concept get split across multiple $\tilde{f}$?
>
> As observed in Figures 3 and 10, the same semantic concept occasionally gets split into multiple masks, particularly in images featuring a single large object in the foreground. However, when objects are split into multiple masks, a consistent pseudo class ID is typically assigned.
>
> ## Q10: For “Further mask proposal”, do you still consider the first principal component and select second, third and so forth similar vectors as spatial anchors? or second, third, forth principal components are used? Please mention this explicitly.
>
> In every iteration, the first principle component of the unassigned features is used. The intuition here is that this is the most dominant representation in the feature distribution of a single image, and we iteratively subtract it from the distribution until the majority of image features is assigned to mask proposals. In this way, there is no need to predefine a fixed number of PCA components, as the number of masks required to segment an image can differ drastically, even in the same dataset. Consider Fig. 1, where the middle image can be decomposed with 3 masks while the top row image has significantly more semantic categories present. We will clarify this in the paper.
>
> ## Q11: As far as I understood, selection of Z is based on how much image region is left to be segmented. In the paper it says 95%, is this threshold based on some statistical distribution of segments within natural images?
>
> This choice is empirical, based on the downstream accuracy on the segmentation task.
> We also ran experiments showcasing the robustness of our method to the stopping criterion for DINO ViT-B/8 unsupervised semantic segmentation on the COCO-Stuff dataset. Although a threshold of 0.9 yields slightly better results in this setup, we believe it is prudent to select a higher threshold to obtain mask proposals for smaller regions, especially in datasets with finely resolved objects (e.g. Cityscapes). However, setting the threshold too high may result in masks containing only a few individual pixels in the final iterations.
>
> |Stop Crit. | 0.85 | 0.9 | 0.95 | 0.99 |
> |-----------|-----:|----:|-----:|-----:|
> |Acc        | 49.2 | 49.3| 48.4 | 47.8 |
> |mIoU       | 21.9 | 22.1| 21.9 | 21.7 |
>
> ## Q 12: Instead of SSL method, how would a supervised, pre-trained method behave? Did authors try that?
>
> The line of research on weakly supervised segmentation -- a very broad field -- addresses exactly this setting.
> Our approach, however, was developed with a completely unsupervised scenario in mind.
> Therefore, it is unlikely to be competitive with weakly supervised methods.
>
> ## Q 13: Selection of K and induced Z? How intricate is their relationship?
>
> We see no immediate connection between K and Z. K is the number of semantic classes we want to discover in the dataset, while Z is the number of mask proposals in a specific image. Z varies from image to image and results from the number of PriMaPs iterations performed until most features have been assigned to the mask, see Sec. 3.1, Deriving PriMaPs.
>
> Repeated references in the related work section. Line with “Nevertheless, most of this work… “
> -- Thank you for pointing this out. We will correct it in the revised version of the paper.

---

> ### Comment · Reviewer_v3pd · 2024-07-22
> **Response to rebuttal**
>
> I thank the authors for addressing my comments and questions. Similar to the other review, my questions were mostly about adding more details and clarification. I would encourage authors to add the mentioned details in the paper, especially the visualizations of the method.
>
> Since the method is novel enough, I believe the community would find this work useful. Therefore, I am inclined to accept this paper.

---

### Author Response · Authors · 2024-07-25
**Paper Revision**

Dear Reviewers,

Thank you once again for your thorough and constructive reviews. We have revised the paper to include your comments and suggestions. All experiments conducted during the rebuttal have been added to the paper.

Changes to the paper are highlighted in blue and the major changes are listed below:
- Extensively revised the method section (Sec. 3.1, Sec. 3.2) as well as Figure 3, incorporated the clarifications provided in the rebuttal, and rewrote parts to make our method easier to understand [v3pd, DBXQ, BQDo].
- Added a paragraph on pseudo-label generation ("PriMaPs pseudo labels" end of Sec. 3.1) after proposal post-processing for a better transition between Sections 3.1 and 3.2 [DBXQ].
- Changed "$\gamma_{psi}$" to "$\gamma$" and "$\gamma_t$" to  "$\kappa$" (Equation 11) [DBXQ].
- Replaced "Class Prototypes $\theta_S$" by "Running Class Prototypes $\theta_R$" and "Momentum Class Prototypes $\theta_T$" by "Momentum Class Prototypes $\theta_M$."
- Changed the term "spatially anchoring" to "anchoring" as it led to confusion.
- Added Figure 2 visualizing the PriMaPs process in more detail [v3pd].
- Added more details on computational requirements to the supplemental (Sec. C.2) [v3pd].
- Added a qualitative comparison of DINOv1 and DINOv2 to the supplemental (Sec B.4 and Fig. 11) [v3pd].
- Added the overclustering experiment to the supplemental (Sec. A.5) [v3pd].
- Added a stop criterion ablation to the supplemental (Sec. A.3 and Fig. 8) [v3pd].
- Added the augmentation breakdown ablation to the supplemental (Sec. A.4 and Tab. 6) [v3pd].
- Pointed out the ablation using the non-iterative implementation of PriMaPs and added the simplified implementation experiment from the rebuttal (Sec. 4.2 -- PriMaPs pseudo-label ablations) [v3pd, DBXQ].
- Pointed more clearly to the threshold ablation experiment (Sec. A.2) in the main paper (Sec. 4.1) [DBXQ].
- Had to move the qualitative PriMaPs examples to the supplemental (Sec. B.5) to stay within the 12-page limit.

We hope you like the revised version of the paper, and we are happy to make further changes if desired.

Thank you very much for your time and effort!

Best regards,

The Authors

---

### Decision · Action_Editor_N173 · 2024-08-09

**Recommendation:** Accept as is

**Comment:**

All three reviewers vote for acceptance. This work has a clear scope at the intersection of self-supervised learning and unsupervised segmentation and shows a clear and simple method for direct inference of masks. The experiments cover timely and relevant methods and datasets, which are also standards for the topic, and therefore can inform the community. This work demonstrates that a simple linear method based on PCA and EM can do well, and without needing to (re-)learn the image representation, and that is worth signaling.

The review process has yielded revisions to the paper that clarify its contents, provide supplemental results and visualizations, and further analyze and ablate the proposed method.

As the action editor, I thank the authors for engaging in the TLMR process and congratulate them on the acceptance of their paper.

**Audience:**

All three reviewers vote for there being an audience at TMLR. The work covers self-supervised learning and its application to segmentation without additional learning by directly extracting masks from the representations. This is relevant to both self-supervised learning, providing a downstream task to evaluate self-supervision for vision, and for vision, in providing a more practical method that does not require additional fine-tuning or optimization. More generally this kind of direct decoding from self-supervised representation to task may be of interest in other modalities for clustering or for segmentation in not just space but time (for instance on audio data). The chosen base models and comparisons are recent and relevant to the machine learning and computer vision community. The work is written clearly for this audience, with appropriate related work, and will be informative to researchers focused on unsupervised segmentation.

**Claims And Evidence:**

All three reviewers vote for agreement between the claims and evidence. This work proposes to derive masks directly from self-supervised representations, by grouping pixels according to prototypes fit by expectation-maximization as principal mask proposals (PriMaPs), and without requiring an addition phase of fine-tuning or learning a new representation. This is indeed simpler, conceptually and computationally, than some existing methods that must optimize an auxiliary representation or train on additional annotations. The experiments cover standard and common base models, providing the self-supervised representations, and benchmarks, providing a variety of segmentations to evaluate, and the experiments that are claimed are delivered. The results are in fact competitive in many cases, and sometimes complement existing methods to further improve results, as when it is combined with Stego or HP.

---

> ### Author Response · Authors · 2024-09-03
> **Camera-ready Uploaded**
>
> We want to thank everyone for the positive decision.  Many thanks to the reviewers and the action editor for all their time, effort, and valuable feedback. We have now uploaded the camera-ready version.